# Sumoylation of the human histone H4 tail inhibits p300-mediated transcription by RNA polymerase II in cellular extracts

Calvin Jon A Leonen[1], Miho Shimada[2†], Caroline E Weller[1], Tomoyoshi Nakadai[2,3], Peter L Hsu[4,5‡], Elizabeth L Tyson[1], Arpit Mishra[6], Patrick MM Shelton[1], Martin Sadilek[1], R David Hawkins[6], Ning Zheng[4,5], Robert G Roeder[2]*, Champak Chatterjee[1]*

[1]Department of Chemistry, University of Washington, Seattle, United States; [2]Laboratory of Biochemistry and Molecular Biology, The Rockefeller University, New York City, United States; [3]Project for Cancer Epigenomics, Cancer Institute of JFCR, Tokyo, Japan; [4]Department of Pharmacology, University of Washington, Seattle, United States; [5]Howard Hughes Medical Institute, University of Washington, Seattle, United States; [6]Department of Genome Sciences, Department of Medicine, University of Washington, Seattle, United States

*For correspondence:
roeder@rockefeller.edu (RGR);
champak1@uw.edu (CC)

Present address: †Department of Molecular Biology, Yokohama City University, Kanagawa, Japan; ‡Department of Structural Biology, Genentech, South San Francisco, United States

Competing interest: The authors declare that no competing interests exist.

**Abstract** The post-translational modification of histones by the small ubiquitin-like modifier (SUMO) protein has been associated with gene regulation, centromeric localization, and double-strand break repair in eukaryotes. Although sumoylation of histone H4 was specifically associated with gene repression, this could not be proven due to the challenge of site-specifically sumoylating H4 in cells. Biochemical crosstalk between SUMO and other histone modifications, such as H4 acetylation and H3 methylation, that are associated with active genes also remains unclear. We addressed these challenges in mechanistic studies using an H4 chemically modified at Lys12 by SUMO-3 (H4K12su) and incorporated into mononucleosomes and chromatinized plasmids for functional studies. Mononucleosome-based assays revealed that H4K12su inhibits transcription-activating H4 tail acetylation by the histone acetyltransferase p300, as well as transcription-associated H3K4 methylation by the extended catalytic module of the Set1/COMPASS (complex of proteins associated with Set1) histone methyltransferase complex. Activator- and p300-dependent in vitro transcription assays with chromatinized plasmids revealed that H4K12su inhibits both H4 tail acetylation and RNA polymerase II-mediated transcription. Finally, cell-based assays with a SUMO-H4 fusion that mimics H4 tail sumoylation confirmed the negative crosstalk between histone sumoylation and acetylation/methylation. Thus, our studies establish the key role for histone sumoylation in gene silencing and its negative biochemical crosstalk with active transcription-associated marks in human cells.

## Editor's evaluation

This manuscript uses chemically synthesized histone H4 with K12 sumoylation coupled with in vitro transcription assay and other in vitro as well as cellular biochemical assays to provide insights about the function of histone H4 K12 sumoylation. The results suggest that H4 K12 sumoylation suppresses H4 acetylation and H3 K4 methylation, two modifications that promote transcription. The biochemistry is very nicely done and the finding may be of interest to the people interested in histone modifications.

## Introduction

Chromatin is Nature's elegant architectural solution to the challenge of packing approximately 3 billion base-pairs of human genomic DNA in an average nuclear volume of only about 500 µm$^3$. Histones constitute the main protein component of chromatin and their reversible post-translational modifications (PTMs), or *marks*, regulate chromatin structure and function by a range of direct and indirect mechanisms (*Kouzarides, 2007*). Based upon their association with either transcriptionally active or silenced regions of chromatin, histone marks were proposed to constitute an epigenetic *code* for gene function (*Strahl and Allis, 2000*). As a consequence of their early discovery and the development of modification-specific chemical and molecular biological tools, marks such as methylation (*Greer and Shi, 2012*), acetylation (*Shahbazian and Grunstein, 2007*), and ubiquitylation (*Weake and Workman, 2008*) have been extensively investigated in vitro and in cell culture. In contrast, histone modification by the small ubiquitin-like modifier (SUMO) protein is a poorly understood mark due both to its very low abundance in cells, which prevents the isolation of sumoylated histones in quantities required for biochemical analysis, and to a lack of sumoylated histone-specific antibodies for cellular studies. First reported in human HEK293T and P493-6 B cells by *Shiio and Eisenman, 2003*, histone sumoylation also occurs in yeast (*Ryu et al., 2019*), parasitic protozoans (*Issar et al., 2008*), and plants (*Miller et al., 2010*). Similar to histone ubiquitylation, sumoylation occurs on all core histones, the linker histone H1, the histone variants H2A.Z and H2A.X, and the centromeric histone variant Cse4 in yeast (*Hendriks and Vertegaal, 2016*; *Ohkuni et al., 2016*). Myriad roles have been proposed for histone sumoylation in different organisms, including transcriptional regulation, kinetochore assembly, the regulation of chromatin structure, and double-strand break repair (*Ryu and Hochstrasser, 2021*). Pioneering efforts to identify specific lysine sites of sumoylation identified K12 in histone H4 as a major recurring site of sumoylation by SUMO-2/3 (H4K12su) (*Galisson et al., 2011*; *Hendriks et al., 2014*), although multiple proximal lysines in the H4 N-terminal tail may also be enzymatically sumoylated in vitro (*Hendriks and Vertegaal, 2016*). Genetic studies in yeast and human cells have typically associated H4 sumoylation with the repression of gene transcription, although mechanistic studies of the direct roles for histone sumoylation in human cells have remained intractable due to the dynamic nature and low abundance of sumoylation (*Shiio and Eisenman, 2003*; *Nathan et al., 2006*).

In an effort to understand the direct effects of H4K12su in chromatin, we previously applied a disulfide-directed chemical sumoylation strategy to generate uniformly and site-specifically sumoylated nucleosome arrays (*Dhall et al., 2014*). Biophysical studies of chromatin-array compaction remarkably showed that H4K12su is incompatible with the compact chromatin structures seen in transcriptionally silent heterochromatin. Subsequent biochemical studies revealed that H4K12su stimulates intranucleosomal activity of the H3K4me2-specific histone demethylase LSD1 (*Dhall et al., 2017*). These studies suggested that sumoylated H4 does not directly enable heterochromatin formation and may instead act by recruiting LSD1 to genes. However, a potentially direct effect of histone H4 sumoylation on promoter-driven transcription by RNA polymerase II (RNAPII) and associated initiation factors that are key for efficient eukaryotic gene transcription has remained unknown.

Pioneering studies of the reconstitution of class II promoter-driven accurate eukaryotic transcription in both nuclear extracts and purified systems have led to insights into roles for histone modifications in gene function (*Dignam et al., 1983*; *Roeder, 2019*). The ability to reconstitute chromatinized plasmid templates using chemically modified histones enables studies of the roles of specific histone modifications in transcription and investigations of their crosstalk with key enzymes associated with transcription initiation and elongation (*Kim et al., 2009*). Multiple proteins involved in gene transcription bind to and modify histone tails, which enables the remodeling of chromatin prior to and during transcription. One such modification, acetylation of lysine side-chains on H3 and H4 by the acetyltransferase p300, is necessary for efficient activator-driven transcription of both 11 nm chromatin (*An et al., 2002*) and 30 nm linker histone H1-containing heterochromatin, likely through mechanisms that include direct decompaction of chromatin upon H4K16 acetylation and octamer eviction by the chromatin remodeler NAP1 (*Robinson et al., 2008*; *Shimada et al., 2019*).

Due to their proposed opposing roles in gene transcription, we investigated the precise nature of biochemical crosstalk between histone sumoylation and histone acetylation by p300. Histone H4 site-specifically sumoylated at Lys12 (H4K12su) was synthesized with the aid of a traceless ligation auxiliary, 2-aminooxyethanethiol (*Weller et al., 2014*), and then incorporated into histone octamers for subsequent reconstitution of cognate mononucleosomes and chromatinized plasmids. Each sumoylated

substrate was subjected to acetyltransferase assays with the full-length p300 enzyme, which revealed the inhibition of acetylation in the H4K12su tail. Consistent with this observation and requirements for both H3 and H4 acetylation for efficient in vitro transcription of chromatin (*An et al., 2002*), replacing wild-type (wt) H4 with H4K12su in chromatinized plasmid templates dramatically inhibited p300-dependent, RNAPII-mediated transcription in vitro. Bottom-up mass spectrometry on chromatinized histones, following a novel in-gel desumoylation protocol, revealed decreased acetylation in H4K12su by p300 when compared to wt H4 acetylation. Consistent with a role in gene repression, H4K12su also inhibited H3K4 methylation by the extended catalytic module (eCM) of the Set1/COMPASS (complex of proteins associated with Set1) methyltransferase complex (*Hsu et al., 2019*). To confirm the negative crosstalk with SUMO in human cells, linear non-hydrolyzable genetic fusions of SUMO-H4 were generated in HEK293T cells and analyzed by western blotting and chromatin immunoprecipitation followed by high-throughput sequencing of the associated DNA (chromatin immunoprecipitation [ChIP]-seq).

Collectively, our observations provide the first unambiguous biochemical demonstration that sumoylated histone H4 directly inhibits RNAPII-mediated transcription from chromatin templates, and reveal its direct negative crosstalk with histone acetylation by p300 and methylation by Set1/COMPASS that are strongly associated with active gene transcription.

## Results

### Reconstitution of site-specifically sumoylated octamers and nucleosomes

Site-specifically sumoylated human histone H4 at Lys12, H4K12su, was obtained by a semisynthetic strategy using the ligation auxiliary 2-aminooxyethanethiol (*Figure 1A and B*; *Dhall et al., 2017*). The semisynthetic sumoylated H4 was incorporated into octamers with purified recombinant human histones H2A, H2B, and H3 (*Figure 1C* and *Figure 1—source data 1*). Nucleosomes were reconstituted from sumoylated octamers using the 147 bp Widom 601 double-stranded DNA (dsDNA) (*Figure 1D*).

### Histone octamer acetylation by p300

We previously showed that H4K12su stimulates activity of the H3K4me1/2 demethylase, LSD1, in the context of an LSD1-CoREST sub-complex (*Dhall et al., 2017*). The stimulation of histone deacetylase (HDAC) activity of the Set3c complex in yeast was also recently proposed for sumoylated histone H2B (*Ryu et al., 2020*). Although the erasure of specific methyl and acetyl marks in the H3 and H4 tails may facilitate the transcriptionally repressed state of chromatin, there remains no information regarding the re-installation of these marks by the corresponding writer enzymes in the presence of H4K12su. Key among the histone acetyltransferases (HAT) is the enzyme p300 that is recruited to chromatin by transcriptional activators and undertakes histone tail acetylation prior to transcription initiation (*An et al., 2002*; *Kraus and Kadonaga, 1998*; *Kundu et al., 2000*). Given its essential role in transcription, we investigated the effect of H4K12su on histone acetylation by p300 prior to and during transcription by RNAPII.

To investigate the direct biochemical crosstalk between H4K12su and acetylation, a western blot-based HAT assay was developed using a sequence-independent pan-acetyllysine antibody to detect lysine acetylation in all four histones (*Figure 1—figure supplement 1A* and *Figure 1—figure supplement 1—source data 1*). In order to effectively compare acetylation of wt H4 with H4K12su and to strictly exclude any acetylation of the surface-exposed lysines in SUMO-3 attached to H4 (*Figure 1—figure supplement 1B*), it was proteolyzed from H4K12su prior to western blot analysis. To this end, after acetylation by p300, the assay product was heat-inactivated at 65°C for 10 min, followed by addition of the purified catalytic domain of human SENP2 containing residues 365–590 (*Figure 1—figure supplement 1C*; *Mikolajczyk et al., 2007*). Heat inactivation precluded p300 activity during desumoylation, and enabled the direct comparison of acetylation status in H4 and H4K12su.

Histone octamers containing wt H4 were first acetylated with full-length p300 immuno-affinity purified from HEK293T cells with an N-terminal FLAG epitope-tag (*Figure 1—figure supplement 1D*; *Chen et al., 2002*). Western blot analysis showed the robust acetylation of all four histones (*Figure 1E* top panel and *Figure 1—figure supplement 1E*). This is consistent with previous in vitro assays that

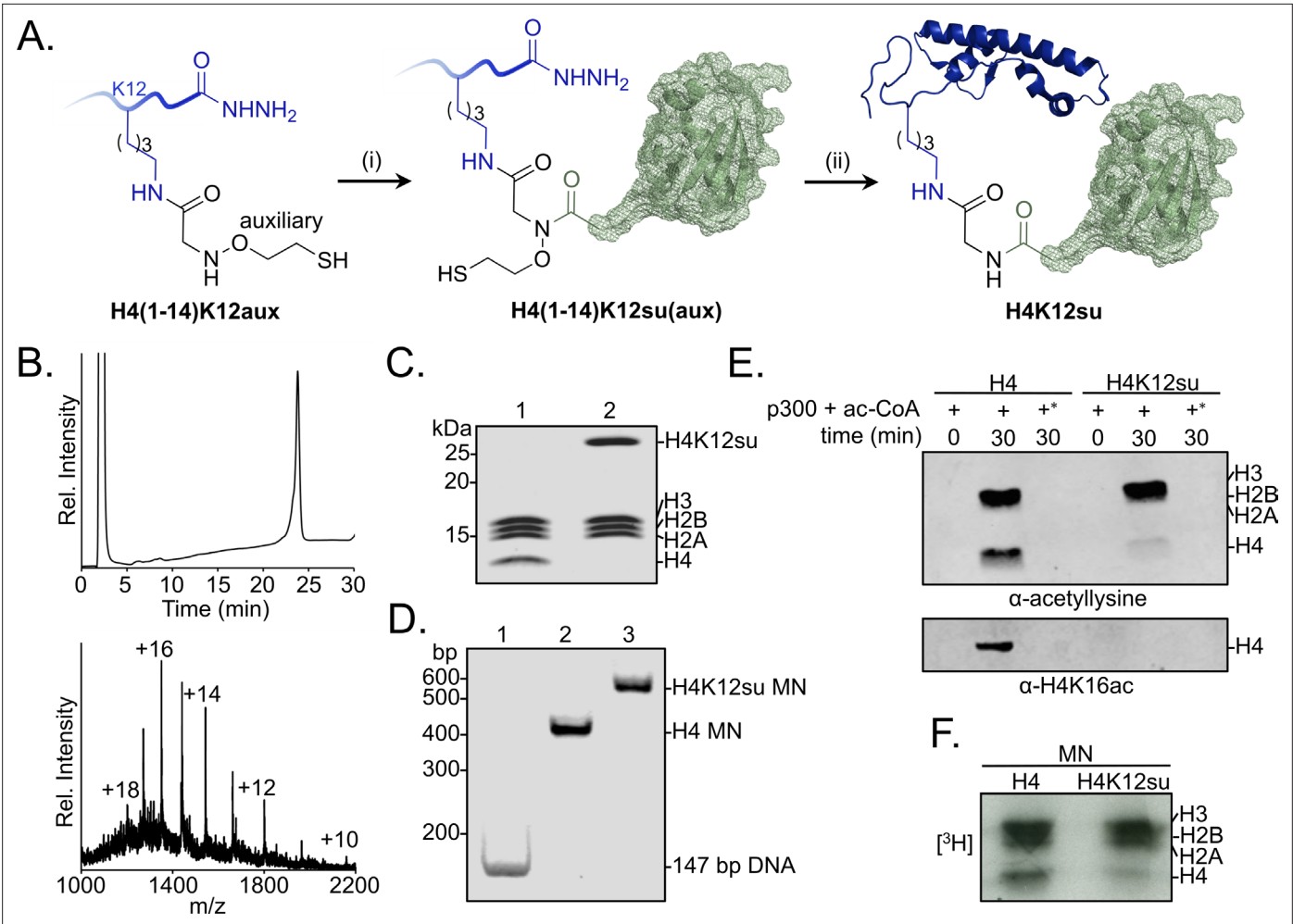

**Figure 1.** Sumoylation inhibits p300-mediated H4 acetylation in octamer and mononucleosome substrates. (**A**) Synthetic scheme for H4K12su. (**i**) An H4(1–14)K12aux peptide was ligated with a SUMO-3 (2–91) C47S α-thioester. (ii) The sumoylated H4(1–14) peptidyl hydrazide containing the auxiliary was converted to a C-terminal α-thioester and ligated with H4(15–102) A15C. The auxiliary was then reductively cleaved from the ligation product. Cys15 in the final ligation product was desulfurized to the native Ala15 to yield site-specifically sumoylated H4K12su. (**B**) C4 analytical RP-HPLC trace of purified H4K12su (top). ESI-MS of purified H4K12su (bottom). Calculated mass 21,596.7 Da. Observed, 21,594.2 ± 3.4 Da. (**C**) Coomassie-stained 15% SDS-PAGE of reconstituted octamers containing wild-type (wt) H4 or H4K12su. (**D**) Ethidium bromide stained 5% TBE gel of mononucleosomes containing wt H4 or H4K12su. (**E**) Western blots of p300 assay products with octamer substrates containing wt H4 or H4K12su, probed with a site-independent pan-acetyllysine antibody (top) and an H4K16ac-specific antibody (bottom). An asterisk indicates assays with heat-inactivated p300 to exclude non-enzymatic acetylation. (**F**) Fluorogram of p300 assay products with [3H]-acetyl-CoA as the co-factor and mononucleosome substrates containing wt H4 or H4K12su.

The online version of this article includes the following figure supplement(s) for figure 1:

**Source data 1.** Unedited intact SDS-PAGE gels for all gel images shown in *Figure 1*.

**Figure supplement 1.** Histone octamer and mononucleosome acetylation by p300.

**Figure supplement 1—source data 1.** Unedited intact SDS-PAGE gels and western blot membranes for all gel and western blot images shown in *Figure 1—figure supplement 1*.

revealed acetylation of all four histones by p300 (*Ogryzko et al., 1996*). Strikingly, H4 from octamers containing H4K12su was devoid of acetylation, including H4K16ac (*Figure 1E* bottom panel), which is strongly associated with chromatin decompaction and active gene transcription (*Robinson et al., 2008*; *Akhtar and Becker, 2000*; *Shogren-Knaak et al., 2006*). Importantly, the inhibition of H4 tail acetylation does not arise from allosteric inactivation of p300, deduced from our observation that H2A/H2B/H3 in H4K12su octamers were acetylated to the same extent as in wt H4 octamers. Additionally, western blots confirmed that SUMO-3 did not inhibit p300 autoacetylation, which is associated with robust acetyltransferase activity (*Figure 1—figure supplement 1F*). Hence the inhibition of

H4 acetylation in H4K12su is likely due to lysine acetylation site occlusion by proximal SUMO-3 in the H4 tail.

## Mononucleosome acetylation by p300

The histone acetylation assay was next undertaken with mononucleosomes containing either wt H4 or H4K12su. We failed to see significant nucleosome acetylation with pan-acetyllysine antibodies with or without pre-incubation of p300 with acetyl-CoA prior to the addition of nucleosomes (*Figure 1—figure supplement 1G*). Due to the significantly decreased activity of p300 with nucleosomal substrates, a [³H]-acetyl-CoA co-factor was employed and the transfer of acetyl groups to histones observed by fluorography. H4 acetylation was also suppressed in mononucleosomes containing H4K12su, but not in unmodified H4 mononucleosomes (*Figure 1F*). Our results unequivocally suggested that SUMO-3 in the H4 tail is inhibitory toward p300-mediated acetylation of chromatin, a process that is necessary for active gene transcription.

Based on the lower acetyltransferase activity observed with mononucleosomes than with octamers, we wondered if dsDNA may inhibit p300 activity. To test this, an equimolar amount of free 147 bp Widom 601 dsDNA was included in the octamer acetylation assay. The presence of DNA was sufficient to inhibit p300 activity to a similar extent as observed with mononucleosomes (*Figure 1—figure supplement 1H*). This unexpected inhibition of p300 activity by free DNA suggests that additional factors, such as transcription factor and RNAPII binding, enable robust p300 activity on histones during transcription initiation and elongation.

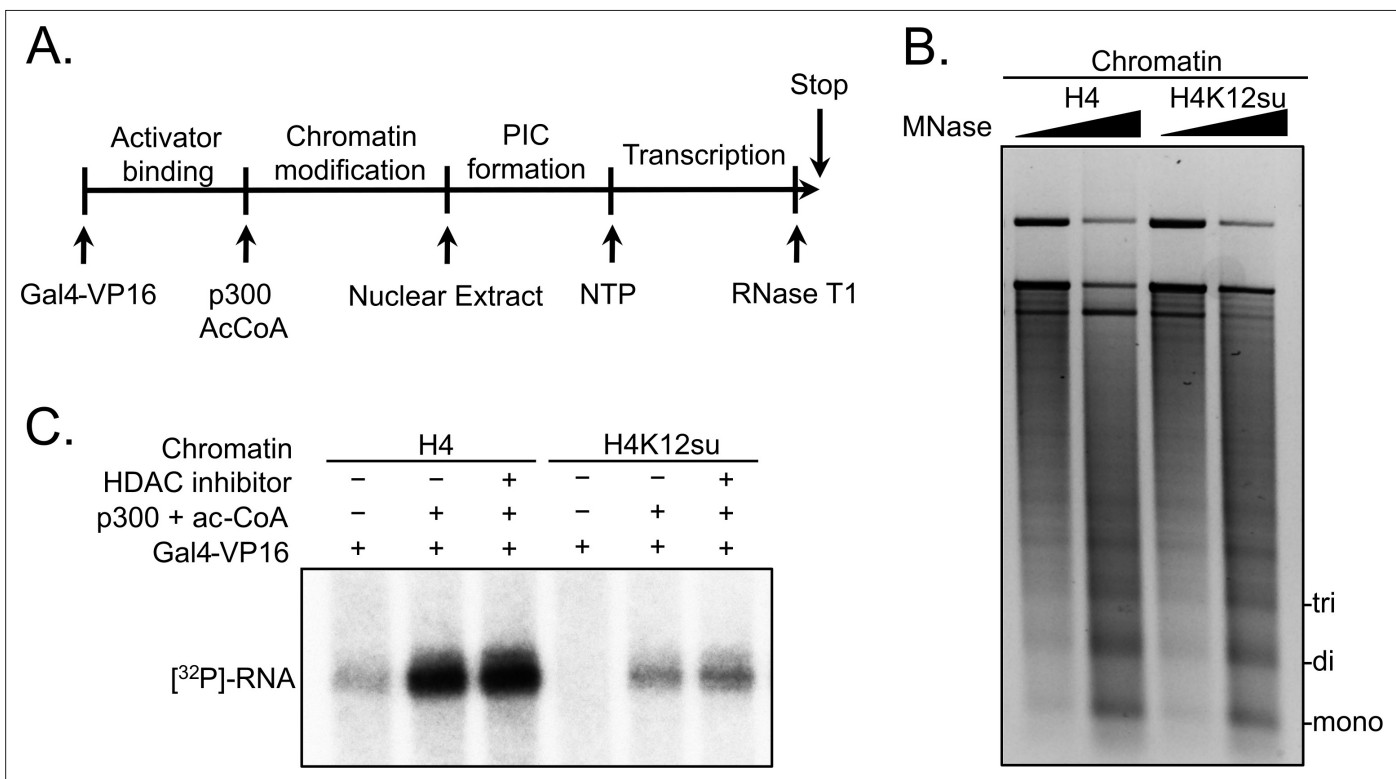

**Figure 2.** Histone H4 sumoylation inhibits in vitro transcription from chromatinized plasmid templates. (**A**) Scheme outlining steps during the in vitro transcription assay with chromatinized plasmids, nuclear extracts, activator Gal4-VP16 and p300. (**B**) Micrococcal nuclease digestion analysis of plasmids chromatinized with wild-type (wt) H4 or H4K12su indicating the similar occupancy and spacing of mononucleosomes. (**C**) Autoradiogram of ³²P-labeled 365 base RNA transcript generated from p300-mediated transcription from chromatinized templates containing wt H4 or H4K12su in the presence or absence of the histone deacetylase (HDAC) inhibitor, trichostatin A.

The online version of this article includes the following figure supplement(s) for figure 2:

**Source data 1.** Unedited intact TBE gel for chromatin digestion gel shown in *Figure 2*.

**Figure supplement 1.** Coomassie-stained SDS-PAGE of chromatin assembly proteins and in vitro transcription components.

**Figure supplement 1—source data 1.** Unedited intact SDS-PAGE gel showing protein components of the in vitro transcription assay.

## The effect of H4K12su on cell-free transcription from chromatinized templates

Based on our observation that H4K12su inhibits the acetylation of key H4 tail residues that are associated with p300-dependent active transcription, including H4K16 that is acetylated in euchromatin, we sought to investigate the direct effect of H4K12su on transcription in our reconstituted cell-free system. These assays employed a p300/Gal4-VP16-dependent transcription system using chromatinized plasmids assembled with reconstituted octamers containing either wt H4 or H4K12su (*Figure 2A*). The plasmid DNA template consisted of five *gal4* binding sites and an ~400 bp G-less cassette (*Kundu et al., 2000*). Due to the absence of any engineered strong nucleosome positioning sequences in the template, chromatinization was undertaken with the histone chaperone NAP1 and the chromatin remodelers Acf1 and ISWI (*Figure 2—figure supplement 1* and *Figure 2—figure supplement 1—source data 1*). Limited micrococcal nuclease (MNase) digestion of the transcription templates revealed the periodic spacing of nucleosomes in chromatin assembled with either wt H4 or H4K12su octamers, and clearly indicated that H4K12su does not inhibit the formation of recombinant chromatin (*Figure 2B* and *Figure 2—source data 1*). In this background, addition of the transcription activator Gal4-VP16, p300, acetyl-CoA, [$\alpha$-$^{32}$P]-CTP, rNTPs, and transcriptional machinery from a HeLa nuclear extract resulted in the transcription of a 365 base RNA from the chromatin template assembled with wt H4 histones (*Figure 2C*). Surprisingly, and in contrast to the direct structural decompaction of chromatin by H4K12su, transcription from templates assembled with H4K12su was drastically inhibited when compared with templates assembled with wt H4. The addition of trichostatin A, a nanomolar inhibitor of class I and II HDACs, did not lead to significant changes in transcription, indicating that the repressive effect of H4K12su is not significantly mediated through HDAC1 in chromatinized templates assembled with non-acetylated histones (*Schultz et al., 2004*). Importantly, our results unambiguously demonstrated transcriptional repression when site-specifically sumoylated H4 was present in chromatin.

## H4 acetylation is inhibited prior to gene transcription in chromatin containing H4K12su

Based on our observations with sumoylated octamer and nucleosome substrates, we wondered if the inhibition of transcription by H4K12su also correlated with diminished H4 tail acetylation by p300. Previous in vitro transcription studies with chromatinized plasmids containing either K-to-R mutations in the H4 tail or truncated H4 missing tail residues 1–19 revealed an ~80% reduced transcriptional output relative to transcription from chromatin containing wt H4 (*An et al., 2002*). Chromatinized plasmids containing either wt H4 or H4K12su were incubated with Gal4-VP16, p300, and acetyl-CoA for 30 min to enable steps preceding transcription; and the histones were subsequently resolved by SDS-PAGE and analyzed by tandem mass spectrometry (MS-MS) after chemical propionylation, trypsination, and separation by capillary-liquid chromatography (*Figure 3—figure supplement 1*; *Sidoli and Garcia, 2017*). A critical innovation in our bottom-up analysis workflow was the use of a SENP2 catalytic domain to desumoylate H4K12su within the polyacrylamide gel matrix after SDS-PAGE. This procedural step was required to generate the same H4(4–17) tryptic peptide from wt H4 and H4K12su after p300-mediated acetylation. The H4(4–17) peptide contains K5,8,12, and 16 that are known to be acetylated by p300 in vitro and in vivo (*Ogryzko et al., 1996*).

Analysis of the H4(4–17) tryptic peptides arising from wt H4 revealed a remarkable degree of hyperacetylation within 30 min. The most abundant peptide corresponded to the K5,8,12,16 tetra-acetylated form with some tri-acetylated species also present (*Figure 3—source data 1*, *Figure 3A*, *Table 1*, and *Figure 3—figure supplements 2 and 3*). This is consistent with the fact that p300 acetylates histones to facilitate transcription (*An et al., 2002*; *Robinson et al., 2008*; *Shimada et al., 2019*; *Kundu et al., 2000*). No significant degree of monoacetylation was observed, and a low abundance of diacetylated peptide was detected after manually searching the MS-MS spectra over the expected elution time-range (*Figure 3—figure supplement 4*). In comparison, chromatin assembled with H4K12su generated significantly fewer hyperacetylated peptides, with approximately equal amounts of tri- and di-acetylated H4(4–17) peptides (*Figure 3B* and *Figure 3—figure supplements 5 and 6*). Small amounts of unmodified H4(4–17) peptides were also observed (*Figure 3—figure supplement 7*). This clearly indicates that H4K12su directly inhibits p300-mediated H4 tail acetylation in the steps

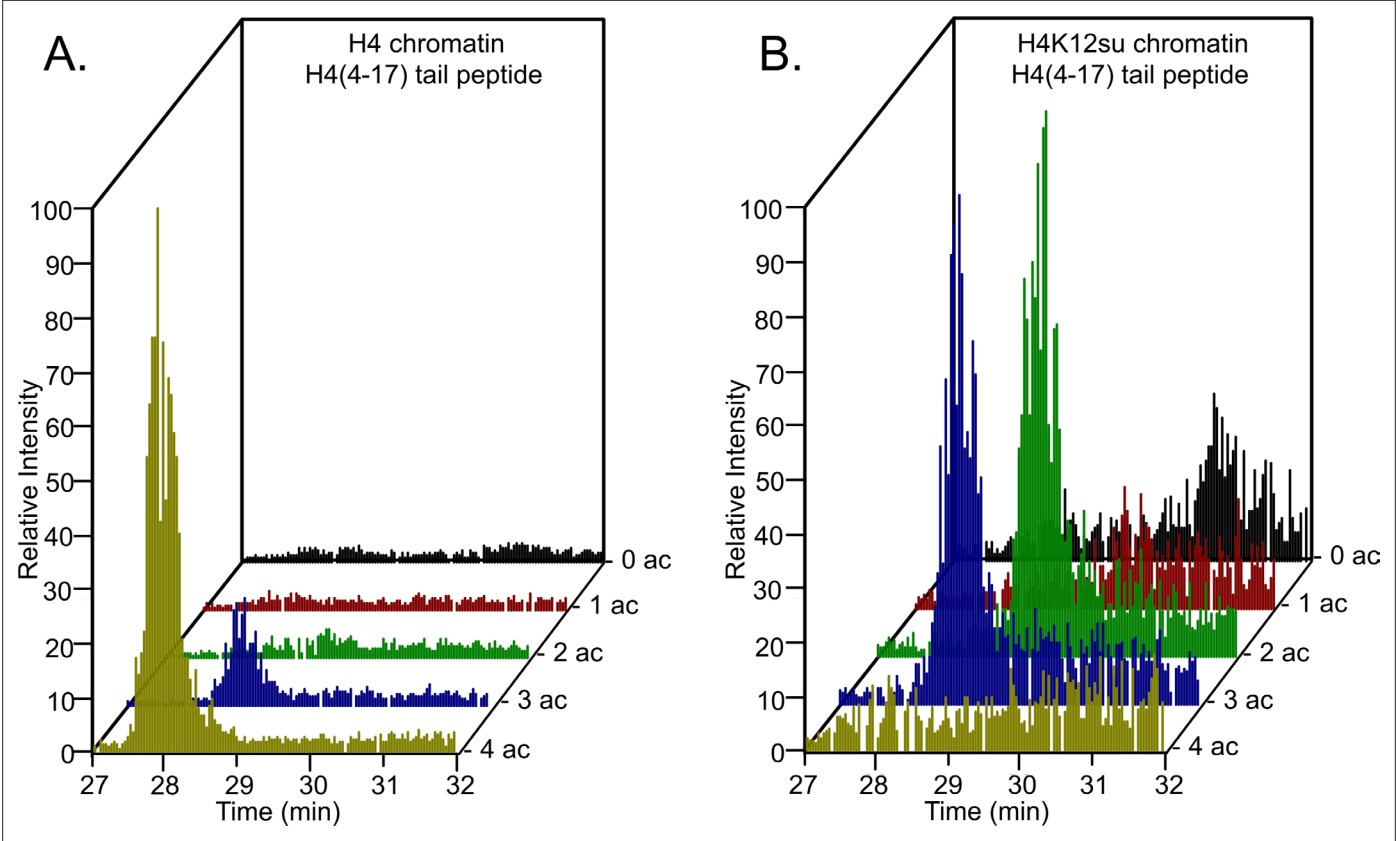

**Figure 3.** Comparison of H4 tail acetylation by p300 in chromatinized plasmid templates with activator Gal4-VP16. (**A**) Extracted ion chromatograms of all H4(4–17) tryptic peptides obtained after SDS-PAGE resolution and in-gel trypsination of acetylated chromatin containing wild-type (wt) H4. (**B**) Extracted ion chromatograms of all H4(4–17) tryptic peptides obtained after SDS-PAGE resolution, in-gel desumoylation and trypsination of acetylated chromatin containing H4K12su. The extracted m/z of each spectrum is centered on the $[M + 2 H]^{2+}$ precursor ion.

The online version of this article includes the following figure supplement(s) for figure 3:

**Source data 1.** Excel file listing all mass spectral data plotted in *Figure 3*, Tables 1–4; and *Figure 3—figure supplements 2–7*.

**Figure supplement 1.** Coomassie-stained SDS-PAGE of histone acetylation assay on chromatinized plasmids containing wild-type (wt) H4 or H4K12su with p300 and activator Gal4-VP16.

**Figure supplement 2.** Tandem MS of tetra-acetylated tryptic peptide H4(4–17).

**Figure supplement 3.** Tandem MS of tri-acetylated tryptic peptide H4(4–17).

**Figure supplement 4.** Tandem MS of di-acetylated tryptic peptide H4(4–17).

**Figure supplement 5.** Representative tandem mass spectrometry (MS-MS) spectrum of tri-acetylated tryptic peptide H4(4–17) generated after in vitro acetylation of chromatinized plasmids containing H4K12su with p300 and activator Gal4-VP16 followed by in-gel desumoylation.

**Figure supplement 6.** Representative tandem mass spectrometry (MS-MS) spectrum of di-acetylated tryptic peptide H4(4–17) generated after in vitro acetylation of chromatinized plasmids containing H4K12su with p300 and activator Gal4-VP16 followed by desumoylation.

**Figure supplement 7.** The tandem mass spectrometry (MS-MS) spectrum of unacetylated tryptic peptide H4(4–17) after in vitro acetylation of chromatinized plasmids containing H4K12su with p300 and activator Gal4-VP16 followed by desumoylation.

prior to transcription. Given the importance of H4 tail acetylation for efficient transcription, H4K12su may inhibit transcription, in part, by directly inhibiting p300 activity on H4.

Since all four acetylated states of H4(4–17) were observed during MS-MS-based analysis of p300 assay products with octameric and chromatin substrates, we interrogated the site specificity of p300 in the H4 tail. Consistent with previous reports, we observed that K5 and K8 are preferred sites in the double acetylated H4 tail, over acetylation at K12 and K16 (*Table 2*; *Schiltz et al., 1999*). Additionally, K12 was preferentially acetylated over K16 in the triply acetylated H4 tail peptide (*Table 3*). These observations were consistent between H4 or H4K12su containing substrates, indicating that the intrinsic substrate preference of p300 is unchanged in the presence of SUMO-3 (*Table 4*).

**Table 1.** H4(4–17) tail peptides acetylated by p300 in chromatinized plasmid templates with activator Gal4-VP16[*,†].

| H4(4–17) peptide | [M] | [M + 2 H]²⁺ | PSM H4 | PSM H4K12su |
|---|---|---|---|---|
| prGKprGGKprGLGKprGGAKprR | 1549.89 | 775.95 | n.d. | 1 |
| prGKprGGKacGLGKprGGAKprR | 1535.88 | 768.95 | n.d. | n.d. |
| prGKacGGKacGLGKprGGAKprR | 1521.86 | 761.94 | 2 | 3 |
| prGKacGGKacGLGKacGGAKprR | 1507.85 | 754.93 | 5 | n.d.[‡] |
| prGKacGGKacGLGKprGGAKacR | 1507.85 | 754.93 | n.d. | 3§ |
| prGKacGGKacGLGKacGGAKacR | 1493.83 | 747.92 | 7 | n.d.[‡] |

[*]Peptides were chemically propionylated before and after trypsinization to cap unmodified lysine side-chains and newly generated N-termini.

[†]Tandem mass spectrometry (MS-MS) spectra observed contained major fragments for the shown modification pattern over other potential patterns, however, no singly acetylated peptides were observed for wild-type (wt) H4.

[‡]Acetylation at K12 is not possible for H4K12su.

[§]The triply acetylated peptide from H4K12su is blocked from acetylation at K12, but is propionylated after in-gel desumoylation. PSM = peptide spectral match. n.d. = not detected.

## Biochemical crosstalk between H4 sumoylation and acetylation in human cells

In order to further probe the biochemical relationship between histone H4 sumoylation and acetylation in human cells, HEK293T cells were transfected with a plasmid encoding an N-terminally HA-tagged *SUMO-3-H4* gene fusion. This fusion positioned SUMO-3 at the N-terminus of H4 proximal to K12 in the tail while also maintaining intact epitopes for antibodies targeting H4K12ac or H4K16ac. To preclude desumoylation by intracellular SENPs, the two C-terminal Gly residues in SUMO-3 were omitted to generate the fusion protein HA-Su3(ΔGG)-H4. After 48 hr of transfection, nuclear chromatin was isolated and treated with MNase to produce mononucleosomes (*Figure 4A*). Only nucleosomes containing either one or two copies of HA-Su3(ΔGG)-H4 were immunoprecipitated using anti-HA agarose beads, and the acetylation state of H4K12 and H4K16 was investigated in western blots using site-specific antibodies (*Figure 4—source data 1*). This analysis revealed that the attachment of SUMO-3 to the N-terminus of H4 significantly inhibits H4K16ac and has a smaller inhibitory effect on H4K12ac (*Figure 4B*). If the K12 side-chain were sumoylated, then K12 would also be completely blocked from acetylation. The presence of all core histones in the pulldown also demonstrated that the ectopically expressed HA-Su3(ΔGG)-H4 fusion was incorporated into chromatin within human cells (*Figure 4—figure supplement 1*). Thus, the biochemical relationship observed in HEK293T cells is consistent with our observation that H4K12su inhibits H4K16ac in vitro, and further indicates a degree of plasticity in the precise site of sumoylation that may exhibit negative biochemical crosstalk with H4K16ac. These cellular studies also expand our discovery that H4 tail sumoylation inhibits p300 activity to include other known H4K16 acetyltransferases such as hMOF (*Taipale et al., 2005*) and CBP (*Bannister and Kouzarides, 1996*).

## H3K4 methylation by COMPASS is inhibited in nucleosomes containing H4K12su

Along with H4K16ac, trimethylation at Lys 4 in histone H3 (H3K4me3) is a key mark associated

**Table 2.** Comparisons of relative ion intensities of characteristic fragment ions from an enzymatically di-acetylated and chemically propionylated H4(4–17) peptide, [M + 2 H]²⁺ = 762 Da, after activator and p300-mediated acetylation of chromatinized plasmids containing wild-type (wt) H4[*].

| Species | Ion | % of Total ion intensity | | Avg. ratio |
|---|---|---|---|---|
| K5ac, K8ac | b₅⁺ | 1.316 | 1.654 | 12.4 |
| K5pr, K8pr | | 0.072 | 0.253 | |
| K12ac, K16ac | y₉⁺ | 0.042 | 0.306 | 0.2 |
| K12pr, K16pr | | 3.176 | 0.971 | |

[*]Only two unique spectra were observed and analyzed for the doubly acetylated and propionylated H4(4–17) tail peptide from wild-type (wt) H4 chromatin.

**Table 3.** Comparisons of relative ion intensities of characteristic fragment ions from an enzymatically tri-acetylated and chemically propionylated H4(4–17) peptide, $[M + 2 H]^{2+}$ = 755 Da, after activator and p300-mediated acetylation of chromatinized plasmids containing wild-type (wt) H4[*].

| Species | Ion | % of Total ion intensity | | | Avg. ratio |
|---|---|---|---|---|---|
| K5ac, K8ac, K12ac | $b_9^+$ | 3.032 | 4.938 | 3.981 | 24.1 ± 13.3 |
| K5ac, K8ac, K12pr | | 0.339 | 0.096 | 0.147 | |
| K5ac, K8ac, K12ac | $b_{10}^+$ | 0.875 | 0.862 | 1.109 | 10.3 ± 5.1 |
| K5ac, K8ac, K12pr | | 0.112 | 0.152 | 0.064 | |
| K12pr, K16ac | $y_8^+$ | 0.271 | n.d. | 0.012 | 0.2 |
| K12ac, K16pr | | 0.885 | 1.005 | 1.263 | |
| K12pr, K16ac | $y_9^+$ | n.d. | 0.123 | 0.054 | 0.02 |
| K12ac, K16pr | | 2.587 | 4.442 | 6.296 | |

[*]Three unique spectra corresponding to the tri-acetylated and propionylated H4(4–17) tail peptide from wild-type (wt) H4 chromatin were analyzed. Error reported is standard deviation of the mean. n.d. = not detected.

with active gene transcription (*Santos-Rosa et al., 2002*). In humans, H3K4me3 is installed by the SETD1A/B and MLL-1/2 enzyme complexes, while the corresponding yeast enzyme complex is COMPASS (*Shilatifard, 2012*). Importantly, the catalytic module (CM) that imparts enzymatic activity and product specificity is evolutionarily conserved in animals and yeast, and consists of Set1 and the subunits Swd1, Bre2, Swd3, and Sdc1 in COMPASS. In an effort to understand the mechanism of auto-regulation in SET1/MLL enzymatic complexes, we recently reported the reconstitution and structural characterization of an extended catalytic module (eCM) of COMPASS that contains both the nSET domain of Set1 and the Spp1 subunit (*Figure 5A*; *Hsu et al., 2019*). Although ubiquitylation at H2BK120 stimulates the methyltransferase activity of the eCM, it is not absolutely critical for nucleosome methylation by SET1/COMPASS complexes in vitro (*Tang et al., 2013*). Based on our previous observation that H4K12su biochemically opposes the presence of H3K4me2 in nucleosomes by stimulating the activity of the H3K4me1/2 demethylase LSD1 (*Dhall et al., 2017*), we asked if H4K12su also directly opposes the installation of H3K4me3 in nucleosomes. Recent cryo-EM structures of the COMPASS eCM bound to the nucleosome core particle show significant spatial separation between the disordered H4 tail and the eCM (*Figure 5A*), thereby making it hard to predict any biochemical crosstalk between sumoylation and methylation (*Hsu et al., 2019*; *Worden et al., 2020*). In order to shed light on this problem, methylation assays were undertaken with mononucleosome substrates

**Table 4.** Comparisons of relative ion intensities of characteristic fragment ions from an enzymatically di-acetylated, desumoylated, and chemically propionylated H4(4–17) peptide, $[M + 2 H]^{2+}$ = 762 Da, after activator and p300-mediated acetylation of chromatinized plasmids containing H4K12su[*].

| Species | Ion | % of Total ion intensity | | | Avg. ratio |
|---|---|---|---|---|---|
| K5ac, K8ac | $b_5^+$ | 2.096 | 2.308 | 1.770 | 71.6 |
| K5pr, K8pr | | n.d. | 0.071 | 0.016 | |
| K5ac, K8ac | $b_6^+$ | 2.603 | 2.835 | 2.900 | 33.1 |
| K5pr, K8pr | | 0.053 | n.d. | 0.170 | |
| K12pr, K16ac | $y_6^+$ | 0.038 | 0.080 | n.d. | 0.04 |
| K12pr, K16pr | | 1.227 | 1.378 | 1.375 | |
| K12pr, K16ac | $y_8^+$ | 0.027 | 0.013 | 0.138 | 0.1 ± 0.09 |
| K12pr, K16pr | | 0.815 | 0.732 | 0.677 | |

[*]Three unique spectra corresponding to the di-acetylated and propionylated H4(4–17) tail peptide from H4K12su chromatin were analyzed. Error reported is standard deviation of the mean. n.d. = not detected.

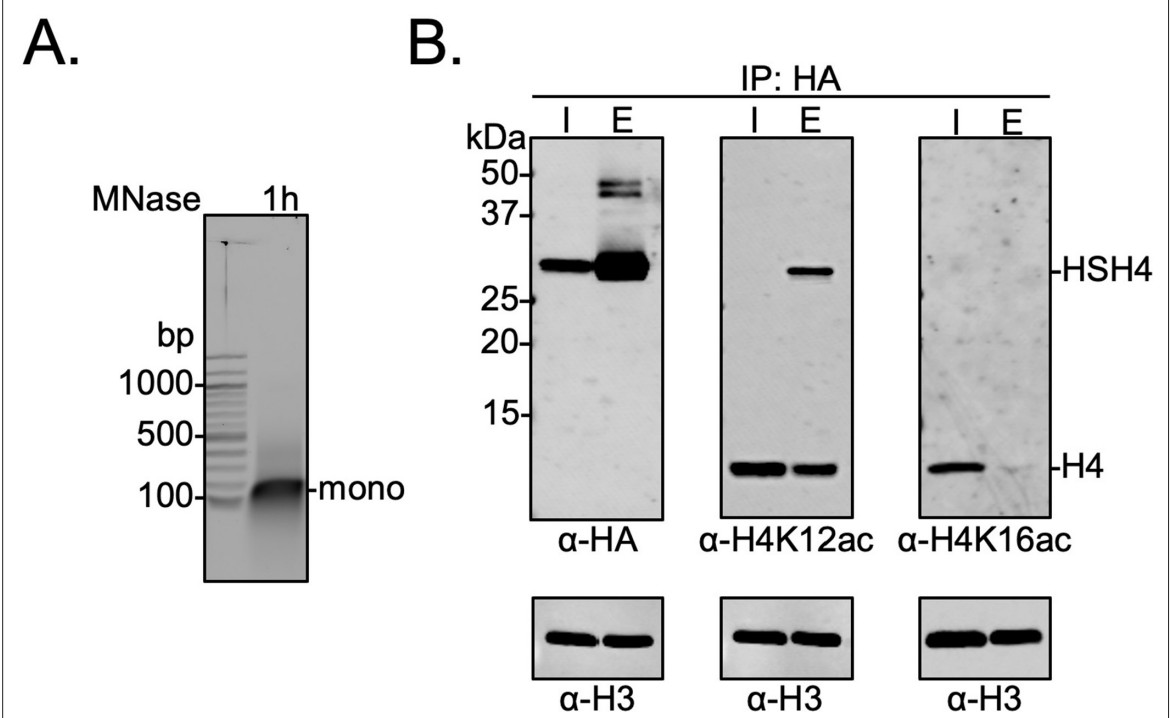

**Figure 4.** Biochemical crosstalk between H4 sumoylation and acetylation in HEK293T cells. (**A**) Extended *microccocal nuclease* digestion of chromatin to generate mononucleosomes that were detected by the presence of ~150 bp DNA in 1.5% agarose gels. (**B**) Immunoprecipitation (IP) from HEK293 cells transfected with HA-Su3(ΔGG)-H4 (HSH4). Input (I) and eluate (E) lanes correspond to undigested bulk chromatin and eluted HA-tagged mononucleosomes containing HSH4. Antibodies targeting H4K12ac and H4K16ac were employed to detect the degree of wild-type (*wt*) H4 and HSH4 acetylation in HA-tagged mononucleosomes. Total histone H3 in each sample was employed as an equal loading control.

The online version of this article includes the following figure supplement(s) for figure 4:

**Source data 1.** Unedited intact SDS-PAGE gels and western blot membranes for all gels and western blot images shown in *Figure 4*.

**Figure supplement 1.** Coomassie-stained SDS-PAGE gel of input (I) and elution (E) samples from immunoprecipitation with anti-HA magnetic beads of *micrococcal nuclease* digested nuclear extracts prepared from HEK293T cells transfected with HA-Su3(ΔGG)-H4.

containing either wt H4 or H4K12su and the six-subunit eCM (*Figure 5—figure supplement 1*). The degree of H3K4me1/2/3 was measured by western blot with antibodies specific for the different H3 methylation states (*Figure 5B* and *Figure 5—source data 1*). These experiments clearly showed that H4K12su inhibits the installation of H3K4me1/2/3 on nucleosomes. Moreover, the negative biochemical crosstalk arises from the presence of the Spp1 subunit in the eCM, because the core five-protein CM (Set1, Swd1, Bre2, Swd3, Sdc1) remained active on nucleosomes with or without the presence of H4K12su (*Figure 5C*). Thus, we conclude that H4K12su in nucleosomes engages in negative biochemical crosstalk with p300-mediated acetylation within the same H4 tail, in cis, and may engage in negative biochemical crosstalk with the COMPASS-mediated methylation in the H3 tail, in trans.

## Biochemical crosstalk between H4 sumoylation and methylation in human cells

In order to probe the biochemical relationship between H4 methylation and sumoylation in HEK293T cells, the degree of H3K4 methylation was investigated in western blots using site- and methylation-state-specific antibodies with immunoprecipitated mononucleosomes containing HA-Su3(ΔGG)-H4. Interestingly, we observed no significant effect on the level of H3K4 trimethylation associated with nucleosomes containing HA-Su3(ΔGG)-H4 relative to total chromatin in the HEK293 chromatin input (*Figure 6A* and *Figure 6—source data 1*). This result clearly indicated that sumoylated H4 does not inhibit all H3K4 methyltransferases in humans. Although Set1/COMPASS is the only H3K4 methyltransferase in yeast, humans have the SETD1A/B and MLL1-4 family of enzymes. The MLL family methyltransferases do not contain analogs of the COMPASS Spp1 subunit that may enforce negative

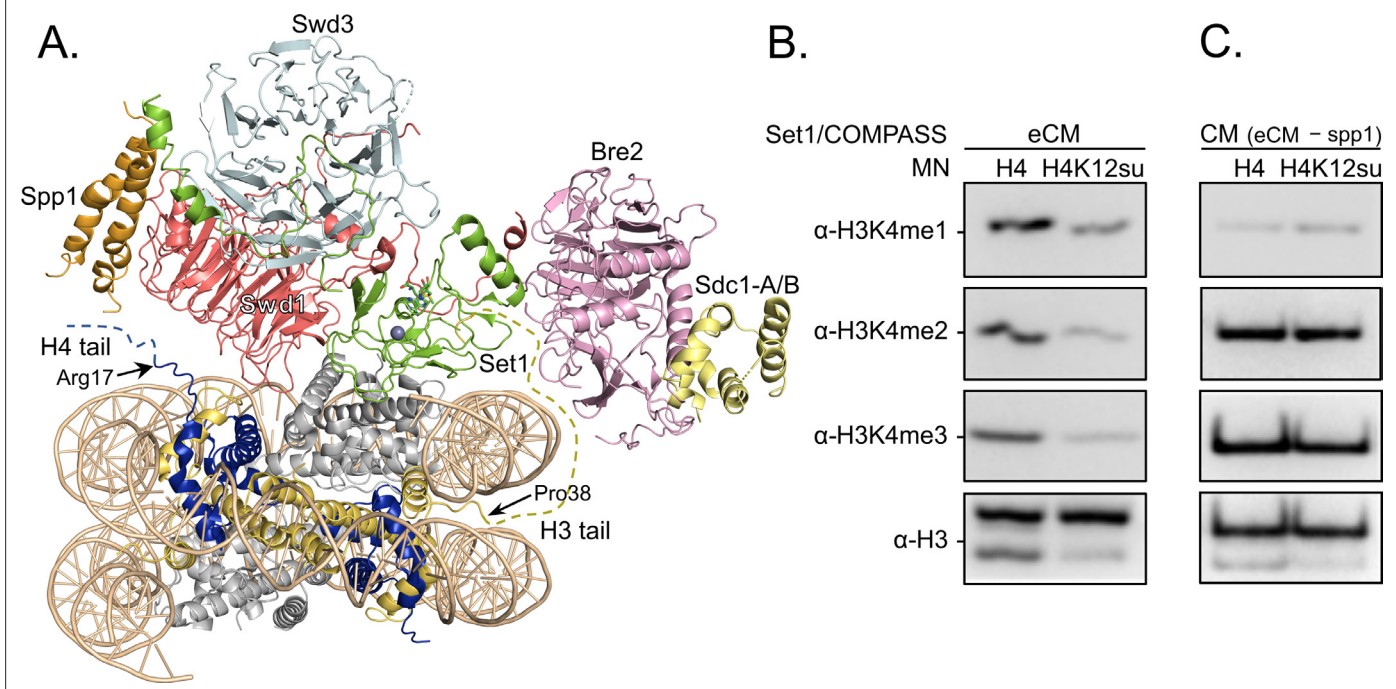

**Figure 5.** H3K4 methylation by the extended catalytic module (eCM) of the complex of proteins associated with Set1 (COMPASS) methyltransferase complex is inhibited by H4 sumoylation. (**A**) Structure of the COMPASS eCM bound to a mononucleosome (PDB code 6UGM). The disordered H3 and H4 tails are shown in gold and blue, respectively, with the last observable amino acid indicated. Dotted lines indicate missing N-terminal amino acids. Spp1 (orange) suppressor of PRP protein 1. Swd3 (light blue), Set1 complex WD40 repeat protein 3. Swd1 (red), Set1 complex WD40 repeat protein 1. Set1 (green), SET domain protein 1. Bre2 (pink), brefeldin-A sensitivity protein 2. Sdc1-A/B (yellow), suppressor of CDC25 protein 1. (**B**) Western blots of the products from methylation assays with mononucleosome substrates containing wild-type (wt) H4 or H4K12su and the COMPASS eCM complex. Mono-, di-, and trimethylated states of H3K4 were detected by the indicated modification-specific antibodies. (**C**) Western blots of the products from methylation assays with mononucleosome substrates containing wt H4 or H4K12su and the COMPASS catalytic module. Mono-, di-, and trimethylated states of H3K4 were detected by the indicated modification-specific antibodies.

The online version of this article includes the following figure supplement(s) for figure 5:

**Source data 1.** Unedited western blot membranes for all western blot images shown in *Figure 5*.

**Figure supplement 1.** Coomassie-stained SDS-PAGE gel of recombinant *Kluyveromyces lactis* complex of proteins associated with Set1 (COMPASS) catalytic module (CM) and COMPASS extended catalytic module (eCM) sub-complexes used in H3K4 methylation assays.

crosstalk with H4 (*Figure 5B*; *Xue et al., 2019*). However, the human SETD1A/B complex has an analog of the Spp1 subunit, namely CXXC-type zinc finger protein 1 (CFP1), which targets SET1 to actively transcribe gene promoters (*Brown et al., 2017*). Therefore, we wondered if the biochemical crosstalk between H4 sumoylation and H3 methylation may be seen in the chromatin occupancy of sumoylated H4 at transcription start sites (TSSs). To address this specific crosstalk, we employed the Flp-In T-REx 293 cell line (Invitrogen) and Flp recombinase-mediated recombination between FRT sites to generate a stable HEK293T cell line containing a single copy of the FLAG-HA-Su3(ΔGG)-H4(Δ1–11) gene under a doxycycline-inducible CMV promoter. In this system, fusion to the N-terminus of H4(Δ1–11) places SUMO-3 at the α-amine of H4K12 and closely mimics enzymatic sumoylation at the ε-amine of H4K12. The addition of doxycycline led to the expression and incorporation of epitope-tagged FLAG-HA-Su3(ΔGG)-H4(Δ1–11) in chromatin as early as 4 hr post-induction (*Figure 6B*). Cells were harvested after 24 hr of doxycycline treatment and regions of genomic DNA associated with FLAG-HA-Su3(ΔGG)-H4(Δ1–11), or with H3K4me3, were identified by ChIP with anti-HA or anti-H3K4me3 antibodies, respectively, followed by high-throughput sequencing of the associated DNA. Our analysis of chromatinized DNA associated with FLAG-HA-Su3(ΔGG)-H4(Δ1–11) and H3K4me3 revealed a striking inverse correlation in their genomic localization (*Figure 6C*). Although H3K4me3 was strongly accumulated at TSSs, FLAG-HA-Su3(ΔGG)-H4(Δ1–11) was diminished at TSSs and enriched at transcription end sites. Thus, ChIP-seq analysis of gene occupancy reiterated the biochemical crosstalk between H4 sumoylation and H3K4me3.

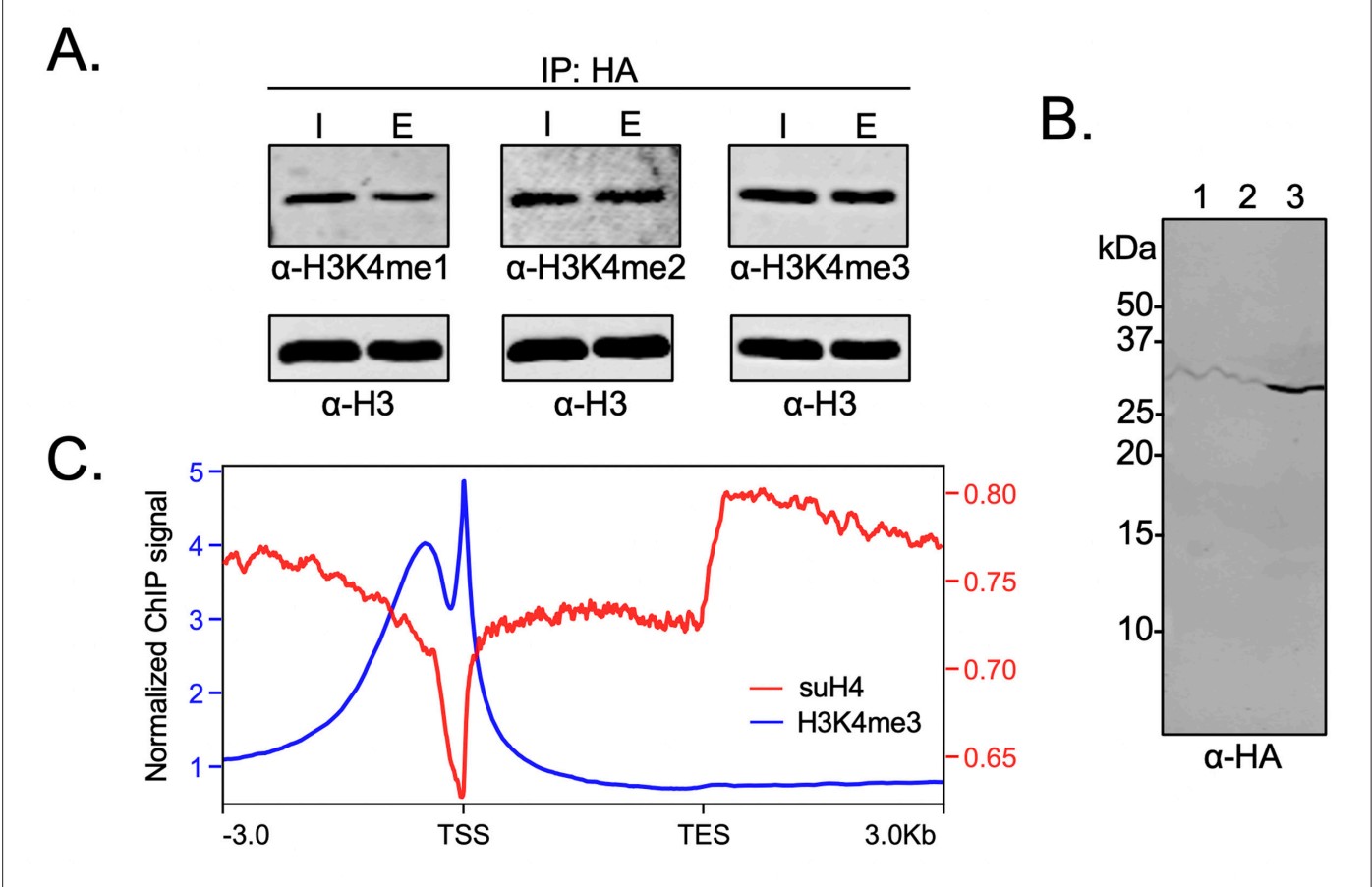

**Figure 6.** Biochemical crosstalk between H4 sumoylation and H3 methylation in HEK293 cells. (**A**) Immunoprecipitation (IP) from HEK293 cells transfected with HA-Su3(ΔGG)-H4 (HSH4). Input (I) and eluate (E) lanes correspond to undigested bulk chromatin and eluted HA-tagged mononucleosomes containing HSH4. Antibodies targeting H3K4me1/2/3 were employed to detect the degree of H3K4 methylation in HA-tagged mononucleosomes. Total histone H3 in each sample was employed as an equal loading control. (**B**) FLAG-HA-Su3(ΔGG)-H4(Δ1–11) (suH4) protein expression in HEK293 cells and its localization to chromatin within 4 hr. Lanes: 1, Cytoplasmic fraction. 2, Nucleoplasmic fraction. 3, Chromatin fraction. (**C**) RPMK normalized chromatin immunoprecipitation (ChIP)-seq signals for SUMO-H4 (red, right y-axis) and H3K4me3 (blue, left y-axis) are plotted across length normalized gene bodies for 19,531 UniProt annotated protein coding genes plus 3 kb upstream of the transcription start site (TSS) and 3 kb downstream of the transcription end site (TES).

The online version of this article includes the following figure supplement(s) for figure 6:

**Source data 1.** Unedited intact gels and western blot membranes for all gels and western blot images shown in *Figure 6*.

## Discussion

Histone marks in eukaryotic chromatin represent a range of biochemical pathways that modulate chromatin structure and function (*Allis et al., 2007*). Marks may directly alter chromatin structure either through their steric bulk or by changing the charge of amino acid side chains. Additionally, marks may recruit chromatin-modifying enzymes that change the modification state of other histones within a nucleosome. The biochemical crosstalk between marks is considered positive when one mark directs installation of another, and negative when one mark opposes the installation of another. The specific sumoylation of histones in chromatin was associated with the repression of gene transcription through early studies in yeast and human cells (*Shiio and Eisenman, 2003*; *Nathan et al., 2006*). We previously discovered the negative biochemical crosstalk between H4 sumoylation and H3K4me2 in nucleosomes mediated by the CoREST-LSD1 sub-complex, which suggested that sumoylation of actively transcribed regions enriched in H3K4me2 may lead to histone demethylation and silencing (*Dhall et al., 2017*). However, the direct effect of H4K12su in chromatin on p300-mediated gene transcription by RNAPII remained entirely unknown. Our semisynthesis of H4K12su using the ligation auxiliary, 2-aminooxyethanethiol, enabled the first interrogation of biochemical crosstalk between histone

sumoylation and p300-catalyzed histone acetylation in reconstituted octamers, nucleosomes, and chromatinized plasmids. We found that although H4K12su has no significant impact on the acetylation of H2A/H2B/H3, in either histone octamer or nucleosomal contexts, it significantly impedes acetylation in the unstructured H4 N-terminal tail. While the simplest explanation is that steric bulk of the 93-amino acid SUMO-3 at K12 prevents acetylation at proximal lysines, this should not be taken for granted as ubiquitin-family modifications at lysines do not always occlude enzymatic activity at proximal sites. For example, the ubiquitylation at K119 in the H2A C-terminus by the polycomb repressive complex 1 (PRC1) E3 ligases Ring1B/Bmi1 does not inhibit ubiquitylation at K124/K127/K129 in the H2A tail by the BRCA1-BARD1 heterodimeric E3 ligase (*Uckelmann et al., 2018*). Reduced acetylation may equally arise from a specific spatial orientation of the H4 tail upon its sumoylation that limits access to the p300 active site.

Given the essential role of acetylated lysines in the H4 tail on chromatin structure and gene transcription in vitro and in cells (*An et al., 2002*), we surmised that diminished H4 acetylation may adversely influence gene transcription. Surprisingly, replacing wt H4 with H4K12su had no significant effect on the efficiency of plasmid chromatinization by the histone chaperone Nap1 and remodelers ACF1 and ISWI, resulting in regularly positioned nucleosomes for both wt H4 and H4K12su. Notably, however, an analysis of these chromatinized plasmid substrates in activator- and p300-dependent transcription assays with nuclear extracts revealed a strong repression of transcription by H4K12su. Toward a further understanding of the H4 acetylation events whose loss may lead to this repression, bottom-up mass spectral analysis with data-independent acquisition of tryptic H4 peptides after SDS-PAGE resolution of the transcription assay components demonstrated reduced acetylation in the H4(4–17) peptide when chromatin was reconstituted with H4K12su. To the best of our knowledge, this is the first mass spectral analysis of acetylation in the tail of chromatin-associated H4 following in vitro transcription, and also the first demonstration that the catalytic domain of the SUMO-specific protease SENP2 can desumoylate histones in SDS-PAGE gels. Given the inherent challenges of detecting SUMO target sites in substrates, due to the lack of a convenient trypsin cleavage site at the C-terminus of SUMO, the ability to selectively remove SUMO using SENP2 may be particularly useful for analyzing sumoylated proteins in complex mixtures that require some degree of separation by SDS-PAGE. As chromatinized plasmids containing H2B ubiquitylated at K120 show similar levels of transcription to unmodified chromatin (*Kim et al., 2009*), the inhibition of transcription we observed may not strictly be due to the steric bulk of SUMO in chromatin. Consistent with this notion, SUMO-3 fused to the N-terminus of H4 in the HA-Su3(ΔGG)-H4 fusion protein also prevented acetylation at K16 in HEK293T cells from as many as 15 residues away. Importantly, the HA-Su3(ΔGG)-H4 fusion protein was readily incorporated into chromatin within 24 hr, as evidenced by the co-immunoprecipitation of all four endogenous histones and ~150 bp DNA after MNase digestion. The presence of endogenous wt H4 in HA-tagged mononucleosomes indicates that HA-Su3(ΔGG)-H4 forms both symmetric and asymmetric nucleosomes in cells. The observation of significantly inhibited acetylation at H4K16 within the HA-Su3(ΔGG)-H4 fusion protein in cells was consistent with our in vitro biochemical observations. Interestingly, endogenous wt H4 in HA-tagged asymmetric mononucleosomes was also deficient in H4K16ac, when compared with H4K16ac in bulk chromatin. This may indicate an additional trans-nucleosome mechanism of acetyltransferase inhibition by SUMO, in addition to its cis-inhibitory effect in the H4 tail.

In addition to histone acetylation, another key histone mark associated with active transcription and enriched at promoter regions is H3K4me3. Installed by the Set1 containing COMPASS complex in yeast and the SET1/MLL1-2 family of methyltransferases in humans, H3K4me3 activates p53- and p300-dependent transcription from chromatinized plasmids (*Tang et al., 2013*). And although H2BK120ub stimulates the methylation of H3K4 on these templates, it is not absolutely critical for SET1 complex activity (*Tang et al., 2013*). Structures of the five-protein core CM of COMPASS (*Hsu et al., 2018*) and the six-protein eCM were recently reported (*Hsu et al., 2019*). While the CM complex does not change methyltransferase activity in the presence of H2BK120ub, the eCM demonstrates some activity on nucleosomes that is further enhanced by the presence of H2BK120ub (*Hsu et al., 2019*). Consistent with this observation, the human SET1 complex also retains some in vitro activity on chromatinized templates lacking H2BK120ub (*Tang et al., 2013*). From the cryo-EM structure of the eCM complex bound to nucleosomes, and the relative position of the unstructured H4 tail (*Figure 5A*), we wondered if H4K12su would have an effect on H3K4me3 methylation by COMPASS.

We discovered that although the CM is not hindered by the presence of H4K12su in nucleosomes, the eCM is significantly hindered by H4K12su. From the differences in composition of the two sub-complexes, we propose that SUMO may sterically interact with the Spp1 subunit in the eCM and may reduce nucleosome binding and/or productive catalysis by the Set1 protein. Our results from methyltransferase assays, in conjunction with previous observations that H4 tail sumoylation inhibits chromatin compaction, appear to indicate that SUMO attached to the H4 tail does not extend away from the nucleosome, but instead may reside on a specific surface of the nucleosome that prevents the close apposition of adjacent nucleosomes and the binding of histone-modifying enzymes . Future structural studies will aim to identify the precise placement of SUMO in sumoylated nucleosomes. As histone acetylation by p300 also stimulates SETD1 activity on chromatin, the direct effect of H4K12su on p300-stimulated SETD1 methylation at H3K4 also remains an interesting question (*Tang et al., 2013*).

Our cellular studies further revealed the complexity of crosstalk between H4K12su and H3K4me3 in humans. Although immunoprecipitated mononucleosomes containing HA-Su3(ΔGG)-H4 did not show a strong negative correlation between SUMO and H3K4me3 across all regions of chromatin, this is consistent with our hypothesis that negative crosstalk is mediated by the Spp1 component of COMPASS (*Figure 5B,C*). The mammalian ortholog of Spp1, namely the CFP1, is entirely absent in the MLL family of H3K4 methyltransferases and is only present in the SETD1A/B family (*Lee and Skalnik, 2005*). Therefore, given the strong association of SETD1A/B with active gene promoters (*Brown et al., 2017*), we undertook ChIP-seq experiments to probe the occupancy of genes in HEK293 cells by H3K4me3 and FLAG-HA-Su3(ΔGG)-H4(Δ1–11). Gratifyingly, we observed a stark negative correlation between gene occupancy by SUMO and H3K4me3 between the transcriptional start site (TSS) and transcription end sites of genes in HEK293 cells. Thus, our extensive investigations of biochemical crosstalk revealed that H4K12su does not inhibit H3K4 methylation genome-wide, but, instead may specifically inhibit the methylation of transcriptionally active genes by the SETD1A/B complexes.

Collectively, the disruption of H4 tail acetylation and H3 tail methylation by the presence of H4K12su along with the inhibition of p300-mediated transcription from chromatinized templates have revealed multiple, albeit not exhaustive, biochemical pathways by which histone sumoylation may inhibit gene transcription (*Figure 7*). These results have shed light on important aspects of chromatin regulation by histone H4 sumoylation and provide a strong mechanistic basis for the proposed roles for SUMO from studies in yeast and cultured human cells.

## Materials and methods

**Key resources table**

| Reagent type (species) or resource | Designation | Source or reference | Identifiers | Additional information |
|---|---|---|---|---|
| Chemical compound, drug | Acetyl-CoA | Roche | 10101893001 | Co-factor for p300 enzyme |
| Commercial assay or kit | Calcium Phosphate Transfection Kit | Thermo | K278001 | Mammalian cell transfection |
| Commercial assay or kit | Lipofectamine 3000 | Invitrogen | L3000001 | Mammalian cell transfection |
| Other | Anti-DYKDDDDK G1 (mouse monoclonal) Affinity Resin | GenScript | L00432 | Antibody-conjugated resin for IP |
| Other | HisPur Ni-NTA resin | Thermo | 88221 | Affinity purification resin |
| Other | Anti-HA (mouse monoclonal) magnetic beads | Pierce | 88836; RRID:AB_2861399 | Antibody-conjugated resin for IP |
| Peptide, recombinant protein | Micrococcal nuclease solution | Thermo | 88216 | Digestion of dsDNA |
| Other | cOmplete, Mini, EDTA-free protease inhibitor cocktail | Roche | 11836170001 | Protease inhibitor cocktail |
| Chemical compound, drug | [$^3$H]-acetyl-CoA | American Radiolabeled Chemicals | ART0213B | Radioactive co-factor for p300 |
| Peptide, recombinant protein | Pierce Trypsin Protease, MS Grade | Thermo | 90057 | Protein cleavage C-terminal to Arg/Lys |

*Continued on next page*

*Continued*

| Reagent type (species) or resource | Designation | Source or reference | Identifiers | Additional information |
|---|---|---|---|---|
| Chemical compound, drug | Propionic anhydride | Sigma-Aldrich | P51478 | Propionylation of peptide lysines and N-terminus |
| Other | DMEM | Gibco | 11956118 | Cell culture medium |
| Other | DPBS | Gibco | 14190250 | Cell culture PBS buffer |
| Other | Fetal bovine serum | Gibco | 16000044 | Cell culture medium additive |
| Other | Amplify fluorographic reagent | GE Amersham | NAMP100 | Tritium decay signal amplifier |
| Other | Kodak GBX developer and fixer | Carestream Health | 1900943 | Immunoblot imaging reagents |
| Chemical compound, drug | Trifluoroacetic acid | Alfa Aesar | AA31771-36 | Peptide synthesis reagent |
| Chemical compound, drug | Formic acid | Acros Organics | AC147932500 | Ion-pairing agent for HPLC |
| Chemical compound, drug | Acetonitrile (ACN) | Fisher | A996 | Solvent for HPLC |
| Other | C18 Zip tip | Millipore | ZTC18S096 | Peptide purification |
| Chemical compound, drug | Glacial acetic acid | Fisher | A38C-212 | Additive for HPLC solvent |
| Strain, strain background (*Escherichia coli*) | *E. coli* BL21(DE3) competent cells | Thermo | FEREC0114 | Chemically competent cells |
| Strain, strain background (*Escherichia coli*) | *E. coli* DH5$\alpha$ competent cells | NEB | C2987HVIAL | Chemically competent cells |
| Cell line (*Homo sapiens*) | HEK 293T | ATCC | CRL-3216; RRID:CVCL_0063 | For transient transfection |
| Cell line (*Homo sapiens*) | Flp-In T-Rex 293 cell line | Invitrogen | R78007 | Stable cell line generation |
| Recombinant DNA reagent | pST100-20xNCP$601$a | Gift from Dr Robert K McGinty | – | Plasmid containing 20 repeats of Widom 601 sequence |
| Recombinant DNA reagent | pcDNA3.1-p300-His$_6$ | Addgene | 23252; RRID:Addgene_23252 | Plasmid for full-length p300 |
| Recombinant DNA reagent | pET28a-His$_6$-SENP2(365–590) | Addgene | 16357; RRID:Addgene_16357 | Plasmid for SUMO protease catalytic domain |
| Recombinant DNA reagent | pcDNA3.1-HA-SUMO-3(ΔGG)-H4 | GenScript | – | This study; generated plasmid containing indicated CDS |
| Recombinant DNA reagent | pcDNA5-FLAG-HA-SUMO-3(ΔGG) H4(Δ1–11) | This study | – | This study; plasmid generated containing indicated CDS |
| Sequence-based reagent | p300_Ctrm_FLAG_R | IDT | – | 5'-ATC CTT GTA ATC GTG TAT GTC TAG TGT ACT C-3' |
| Sequence-based reagent | p300_Ctrm_FLAG_F | IDT | – | 5'-GAT GAC GAT AAA TAG TGA TAC TAA GCT TAA GTT TAA AC-3' |
| Antibody | Rabbit polyclonal anti-acetyllysine antibody | Millipore | AB3879; RRID:AB_11214410 | WB (1:2000) dilution |
| Antibody | Rabbit polyclonal anti-H4K16ac antibody | Active Motif | 39167; RRID:AB_2636968 | WB (1:2000) dilution |
| Antibody | Rabbit polyclonal anti-H4K12ac antibody | Active Motif | 39066 | WB (1:2000) dilution |
| Antibody | Rabbit monoclonal anti-H3K4me1 | Cell Signaling Technology | 5326; RRID:AB_10695148 | WB (1:2000) dilution |
| Antibody | Rabbit polyclonal anti-H3K4me2 | Abcam | ab7766; RRID:AB_2560996 | WB (1:2000) dilution |
| Antibody | Rabbit polyclonal anti-H3K4me3 | Abcam | ab8580; RRID:AB_306649 | WB 1:2000 dilution |

*Continued on next page*

*Continued*

| Reagent type (species) or resource | Designation | Source or reference | Identifiers | Additional information |
|---|---|---|---|---|
| Antibody | Rabbit polyclonal anti-Histone H3 | Abcam | ab1791; RRID:AB_302613 | WB (1:2000) dilution |
| Antibody | Mouse monoclonal anti-Histone H3 | Abcam | ab24834; RRID:AB_470335 | WB (1:2000) dilution |
| Antibody | Rabbit monoclonal anti-HA | Cell Signaling Technology | 3724; RRID:AB_1549585 | WB (1:2000) dilution |
| Antibody | Mouse monoclonal anti-FLAG | Sigma-Aldrich | F1804; RRID:AB_262044 | WB (1:2000) dilution |
| Antibody | Anti-rabbit monoclonal, HRP conjugated | GE Healthcare | NA934; RRID:AB_2722659 | WB (1:40000) dilution |
| Antibody | IRDye 680RD Goat polyclonal anti-Rabbit IgG | Li-COR Biosciences | 926–68071; RRID:AB_10956166 | WB (1:15000) dilution |
| Antibody | IRDye 800CW Goat polyclonal anti-Rabbit IgG | Li-COR Biosciences | 926–32211; RRID:AB_621843 | WB (1:15000) dilution |
| Antibody | IRDye 800CW Goat polyclonal anti-Mouse IgG | Li-COR Biosciences | 926–32210; RRID:AB_621842 | WB (1:15000) dilution |

## HPLC purification

Proteins and peptides were analyzed (4.6 mm × 150 mm, 5 µm) and purified (22 mm × 250 mm, 15–20 µm) with C4 and C18 reverse-phase HPLC columns from Vydac (Deerfiled, IL) on either a Varian Prostar (Palo Alto, CA) or Agilent (Santa Clara, CA) 1260 Infinity II LC system. The mobile phase consisted of Buffer A (0.1% trifluoracetic acid in water), and Buffer B (90% acetonitrile in water, 0.1% trifluoroacetic acid). UV-vis profiles of eluting peptides/proteins were monitored at 214 and 280 nm.

## Electrospray ionization mass spectrometry

Routine peptide/protein mass spectrometry was performed by direct infusion on a Bruker (Billerica, MA) Esquire ion-trap mass spectrometer operating in positive mode.

## Recombinant human histone purification

Human histones H2A 2A, H2B 1K, H3.2, and H4 were expressed from pET3a plasmids in *E. coli* BL21(DE3) cells (*Dhall et al., 2014*). The insoluble histones were extracted from exclusion bodies with 6 M Gn-HCl, 10 mM Tris, pH 7.5. Histones were precipitated by dialysis against Millipore water in Spectra/Por 6 3.5 kDa molecular weight cut-off dialysis tubing and lyophilized to dryness. Crude histones were dissolved in 6 M Gn-HCl and purified by preparative C4 RP-HPLC.

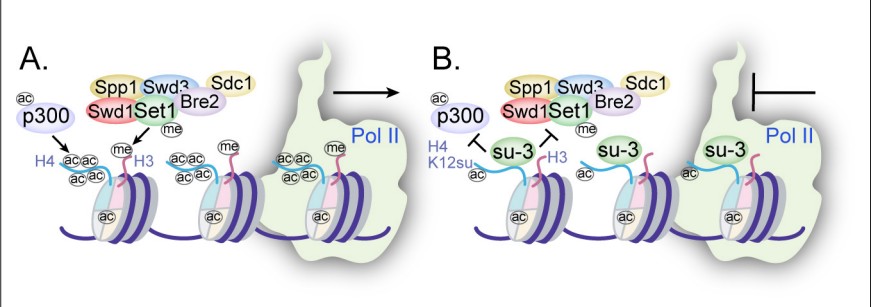

**Figure 7.** Mechanisms of chromatin regulation by H4K12su. (**A**) Transcription from chromatinized templates containing wild-type (wt) H4 is accompanied by acetylation of all four histones by p300, and with the methylation of the H3 tail by the complex of proteins associated with Set1 (COMPASS)/SET1 complexes. (**B**) The inhibition of PolII-mediated transcription from chromatinized templates containing H4K12su is accompanied by reduced p300-mediated H4 tail acetylation. H4K12su also inhibits H3 tail methylation by the extended catalytic module of the COMPASS complex. For clarity, only one of two histone tails, each, is shown for H3 and H4.

## Semisynthesis of H4K12su

The peptide H4(1–14)K12ivDde-C(O)NHNH$_2$ was synthesized on 2-chlorotrityl chloride resin by standard 9-fluorenylmethoxycarbonyl (Fmoc)-based solid phase peptide synthesis on a CEM Liberty Blue Automated Microwave Peptide Synthesizer (Matthews, NC) (*Dhall et al., 2017*). The protected ligation auxiliary, *O*-(2-(tritylthio)ethyl)hydroxylamine, was incorporated at K12 by coupling bromo-acetic acid at deprotected K12 on the resin followed by displacement of the bromide with 0.25 M auxiliary in dry DMSO. After acidolytic cleavage from the resin and C18 RP-HPLC purification, the H4(1–14)K12aux-C(O)NHNH$_2$ peptide (six equivalents) was ligated with purified SUMO-3 (2–91) C47S-MESNa thioester (one equivalent) in ligation buffer consisting of 6 M Gn-HCl, 100 mM Na$_2$HPO$_4$, pH 7.3 to generate H4(1–14)K12su(aux)-C(O)NHNH$_2$ after 24 hr at 25°C. After C18 RP-HPLC purification, the sumoylated H4(1–14) peptidyl hydrazide was converted to the C-terminal acyl azide by diazotization with 15 equivalents of NaNO$_2$ in 200 mM Na$_2$HPO$_4$, 6 M Gn-HCl, pH 3.0 at –20°C for 20 min. The acyl azide was converted to the 4-mercaptophenylacetic acid (MPAA) C-terminal α-thioester in situ and ligated with purified H4(15–102) A15C truncant protein (two equivalents) in a ligation buffer consisting of 200 mM Na$_2$HPO$_4$, 6 M Gn-HCl, 200 mM MPAA, pH 6.5. The pH was adjusted to 6.8–7.0 and ligation allowed to proceed at 25°C for 24 hr. The full-length H4(A15C)K12su(aux) ligation product was purified by C4 RP-HPLC and the ligation auxiliary subsequently cleaved by activated Zn in 6 M Gn-HCl, pH 3.0, at 37°C under argon over 24 hr. This yielded pure H4(A15C)K12su after C4 RP-HPLC purification. Finally, radical-mediated desulfuriza-tion of Cys15 to the native Ala15 in histone H4 was accomplished by first dissolving H4(A15C)K12su in 100 mM Na$_2$HPO$_4$, 6 M Gn-HCl, 500 mM TCEP, 100 mM MESNa, pH 7.5. To this solution was added 2-methyl-2-propanethiol to a concentration of 280 mM and the radical initiator 2,2′-azobis[2-(2-imidazolin-2-yl)propane]dihydrochloride (VA-044) to a concentration of 10 mM. Desulfurization was allowed to proceed at 37°C for 24 hr and the final sumoylated histone product, H4K12su, was purified by C4 RP-HPLC.

## 147 bp Widom 601 DNA preparation

A plasmid containing 20 repeats of the 147 bp Widom 601 sequence, pST100-20xNCP601a was a kind gift from Prof. Robert K McGinty (UNC, Chapel Hill, NC) (*Dyer et al., 2004*). The plasmid was used to transform *E. coli* DH5α and propagated in LB media. The plasmid was extracted from cells by alkaline cell lysis and precipitated with isopropanol. The precipitate was collected by centrifugation and washed with 70% ethanol, followed by resuspension in 20 mM Tris, pH 8.0. The plasmid was purified by SOURCE 15Q 4.6/100 PE strong anion exchange column and analyzed by 1% agarose gel electrophoresis. Fractions pure from RNA contamination were pooled and dialyzed against 10 mM Tris, pH 8, overnight. The plasmid was cleaved by EcoRV to liberate the 147 bp 601 Widom sequence overnight. The plasmid backbone was separated by PEG precipitation followed by SOURCE 15Q 4.6/100 PE anion exchange. Pure fractions of 147 bp dsDNA, as seen by 1.5% agarose gel electro-phoresis, were pooled and dialyzed against 10 mM Tris, pH 7.0, overnight and stored frozen at –20°C.

## Octamer and mononucleosome formation

All four histones were combined in equimolar amounts in 7 M Gn-HCl, 20 mM Tris, pH 7.5 at a final concentration of 1 mg/mL and dialyzed against 10 mM Tris, 2 M NaCl, 1 mM EDTA, pH 7.5 over-night at 4°C. The self-assembled crude octamers were purified by size exclusion chromatography on a Superdex-200 10/300 GL column attached to an AKTA FPLC (GE Healthcare, Chicago, IL). Pure fractions were identified by 15% SDS-PAGE after Coomassie staining. Pure octamer fractions were combined and concentrated, diluted with 10% glycerol and flash-frozen for long-term storage. Mononucleosomes were formed by mixing equivalent amounts of histone octamers and 147 bp Widom *601* DNA in a buffer consisting of 2 M NaCl and 10 mM Tris, pH 7.0, followed by dialysis against 10 mM Tris to a final NaCl concentration of 200 mM. Small-scale mononucleosome formation tests were first undertaken to fine-tune the molar ratio of octamer to 147 bp DNA before large-scale preparations. This avoided the presence of unbound 147 bp DNA in the mononucleosome prepara-tions as seen by 5% TBE gels stained with ethidium bromide. Freshly prepared mononucleosomes containing wt H4 or H4K12su were stored on ice at 4°C for a maximum period of 3 weeks (*Dhall et al., 2014*).

## Purification of the SENP2 catalytic domain

A pET28a plasmid containing the SENP2 catalytic domain (cat.SENP2) was obtained from Addgene (Catalog number 16357) (*Reverter and Lima, 2009*). The protein was expressed and purified from *E. coli* (DE3) cells. Briefly, cells were grown in 2× YT medium at 37°C until $OD_{600}$ 0.6–0.8 and protein expression was induced with 0.5 mM isopropyl-β-D-galactopyranoside for 3 hr at 25°C. Cells were collected by centrifugation and lysed by sonication in 20% (w/v) sucrose, 20 mM Tris, 350 mM NaCl, 20 mM imidazole, 1 mM β-mercaptoethanol, 1 mM phenylmethylsulfonyl fluoride (PMSF), pH 8.0. The lysate was clarified by centrifugation and bound to $Ni^{2+}$-NTA resin over 1 hr with continuous nutation at 4°C. The resin was then washed with a buffer containing 20 mM Tris, 350 mM NaCl, pH 8.0, followed by successive buffers containing increasing amounts of imidazole from 20 to 100 mM to elute non-specifically bound proteins. Finally, the cat.SENP2 was eluted with a buffer containing 20 mM Tris, 350 mM NaCl, 400 mM imidazole, pH 8.0. Fractions containing cat.SENP2 were identified by 15% SDS-PAGE, pooled and dialyzed in Spectra/Por six dialysis tubing with 15 kDa molecular weight cut-off against 2 L of dialysis buffer containing 20 mM Tris, 100 mM NaCl, 5% (v/v) glycerol, pH 8.0 at 4°C for 2 hr, twice. The dialyzed protein was stored at –80°C in aliquots flash-frozen in liquid nitrogen.

## DNA cloning and sequencing

A Q5 site-directed mutagenesis kit (NEB, Ipswitch, MA) was used to generate pcDNA3.1-p300-FLAG from the plasmid pcDNA3.1-p300-$His_6$ obtained from Addgene (Catalog number 23252) by following the manufacturer's protocols. Oligos for molecular cloning were purchased from Integrated DNA Technologies (Coralville, IA) and DNA sequencing was carried out by Eurofins Genomics (Louisville, KY). The pcDNA3.1-HA-SUMO-3(ΔGG)-H4 was commercially prepared by GenScript (Piscataway, NJ).

## Preparation of pcDNA for transfection

DNA for transient transfection was prepared by Miraprep of *E. coli* DH5α cells as previously described using a Qiagen (Valencia, CA) DNA miniprep kit (*Pronobis et al., 2016*). Briefly, transformed *E. coli* DH5α cells were grown in 50 mL LB media supplemented with ampicillin (50 µg/mL) overnight at 37°C. Cells were collected by centrifugation and resuspended in P1 buffer supplemented with fresh RNase. After alkaline lysis and neutralization, the supernatant was cleared by centrifugation. The supernatant was diluted with an equal volume of 96% (v/v) ethanol prior to loading onto five Qiagen miniprep spin columns. At this point, the DNA was washed and eluted according to the Qiagen protocol. Purity of the eluted DNA was checked by measuring the $A_{280\,nm}/A_{260\,nm}$ ratio on a NanoDrop 2000c spectrophotometer and by agarose gel electrophoresis. The correct gene sequence for p300 was also confirmed by sequencing prior to transfection in human cells.

## Cell lines

The standard HEK293T cells used in this study were purchased from either ATCC or Thermo Fisher Scientific and authenticated by STR profiling. Cell lines tested negative for mycoplasma contamination by the PCR-based Universal Mycoplasma Detection Kit from ATCC.

## HEK293T cell culture

HEK293T cells were cultured in T75 flasks using Dulbecco's modified Eagle medium supplemented with 10% fetal bovine serum and incubated at 37°C in a 5% $CO_2$ atmosphere.

## Transient transfection of HEK293T cells

HEK293T cells were cultured to ~60% confluency before transient transfection with pcDNA3.1-p300-FLAG. The cell growth medium was changed at least 1 hr before transfection. Cells were transfected with the Calcium Phosphate Transfection Kit (Thermo Fisher Scientific, Waltham, MA) using 15 µg DNA per T75 flask for FLAG-p300. Lipofectamine 3000 (Invitrogen) was used to transfect cells with a plasmid containing the *HA-Su3(ΔGG)-H4* gene, and the media was changed 24 hr after transfection. Cells were grown in transfection medium over 48 hr before detachment by trypsin and collection by centrifugation. The cell pellet was washed thrice with ice-cold Dulbecco's phosphate-buffered saline before either protein purification or storage at –80°C.

## Generation of stable cell lines

FLAG-HA-Su3(ΔGG)-H4(Δ1–11) was sub-cloned by Gibson assembly from DNA fragments into the AflII site of the pcDNA5/FRT/TO vector (Invitrogen) to stably introduce these genes into the genome of HEK293 Flp-In TREx cells under the control of a tetracycline (Tet)-inducible promoter. All plasmids were verified by DNA sequencing. Transient transfection and selection of stable transfectants were performed according to the manufacturer's recommendations. Verification of Tet-inducible expression of the stably incorporated gene was confirmed by immunoblotting for the HA-tagged protein in cell lysates.

## Purification of the full-Length HAT P300

The washed HEK293T cell pellet was resuspended in lysis buffer consisting of 20 mM Tris, pH 7.4 at 4°C, 0.5 M KCl, 5 mM MgCl$_2$, 0.1% (v/v) IGEPAL CA-630, 1 mM PMSF, 1× EDTA-free protease inhibitor (Roche, Basel, CH), 10% (v/v) glycerol then frozen in dry ice and allowed to thaw at room temperature. The lysate was nutated for 30 min at 4°C before clarifying by centrifugation at 17620*rcf* for 30 min at 4°C. The soluble fraction was bound to anti-DYKDDDDK resin (Genscript, Piscataway, NJ) for 1 hr at 4°C, then washed four times in lysis buffer and finally once in lysis buffer containing 150 mM KCl without protease inhibitors. The full-length p300-FLAG protein was eluted with 200 µg/mL 3xFLAG peptide in 150 mM KCl containing lysis buffer. The purity and concentration of p300-FLAG were assessed by 8% SDS-PAGE using BSA standards (Thermo Fisher) and by western blot with an anti-FLAG antibody (Sigma). Aliquots of p300-FLAG in 10% (v/v) glycerol containing elution buffer were flash-frozen in liquid nitrogen and stored at –80°C.

## Histone acetylation assays

Histone acetylation assays were conducted as outlined in *Figure 1—figure supplement 1A*. In brief, 30 µL volume assays were prepared by mixing histone octamers (300 nM), acetyl-CoA (25 µM), and p300-FLAG (150 pM) in HAT buffer consisting of 50 mM Tris, pH 8.0, 1 mM DTT, 1 mM PMSF, 0.1 mM EDTA, and 10% (v/v) glycerol. Acetylation assays with octamer substrates were undertaken at 30°C for 30 min and stopped by heating the assay mixture to 65°C for 10 min which denatures p300. For fluorography, 389 nM mononucleosomes containing either H4 or H4K12su were incubated with 2 µL [$^3$H]-acetyl-CoA (15.95 Ci/mmol) and p300 (292 pM) overnight at 30°C before heat denaturation at 65°C for 10 min (*Qiu et al., 2006*). Heat denatured samples were cooled down to 30°C before the addition of cat.SENP2 (three equivalents relative to H4K12su). To ensure similar sample handling, cat.SENP2 was added to both H4 and H4K12su assay samples. After 2 hr of desumoylation with cat. SENP2, the samples were denatured with 6× Laemmli and resolved by SDS-PAGE prior to analysis by western blotting or by fluorography.

## Native ChIP of mononucleosomes

Nuclei were collected from pelleted HEK293T cells transfected with HA-Su3(ΔGG)-H4 by resuspension in a modified STM-N/D buffer (250 mM sucrose, 50 mM Tris, pH 7.4 at 4°C, 5 mM MgCl$_2$, 10 mM iodoacetamide (IA), 20 mM N-ethylmaleimide (NEM), 0.5% IGEPAL CA-630, 1× cOmplete protease inhibitor cocktail, 0.5% (w/v) sodium deoxycholate). Suspensions were incubated on ice for 5 min and then spun down at 25 kRCF for 5 min at 4°C in order to pellet nuclei. The cytoplasmic fraction was removed, and the nuclei were resuspended in Buffer 150 consisting of 20 mM Tris, pH 7.4 at 4°C, 0.15 M NaCl, 5 mM MgCl$_2$, 5 mM CaCl$_2$, 0.1% IGEPAL CA-630, 1 mM PMSF, 10 mM IA, 20 mM NEM, 1× cOmplete protease inhibitor cocktail, and 10% (v/v) glycerol. The nuclei were digested with MNase diluted in 50 mM Tris, pH 8.0 at 37°C, 5 mM CaCl$_2$, 20 mM NEM, 10 mM IA, 0.1% (v/v) Triton X-100, 1× cOmplete cocktail. Chromatin digestion was allowed to proceed for 1 hr in a 37°C water-bath, and the sample was gently mixed every 30 min. Digestion was halted by the addition of 0.5 M EGTA and the solution was passed through a 25 G needle five times to reduce its viscosity. The resulting solution was diluted with one volume of Buffer 500 (20 mM Tris, pH 7.4 at 4°C, 0. 5 M NaCl, 5 mM MgCl$_2$, 5 mM CaCl$_2$, 0.1% IGEPAL CA-630, 1 mM PMSF, 10 mM IA, 20 mM NEM, 1× cOmplete cocktail, 10% (v/v) glycerol) and then supplemented with additional salt to a final salt concentration of 0.5 M NaCl. The digested chromatin was clarified by centrifugation at 17.2 kRCF at 4°C for 20 min, and the soluble fraction was collected. A sample of the soluble fraction was mixed with 1% (w/v) SDS and analyzed by 1.2% agarose gel electrophoresis to check for mononucleosomes by the presence of ~150 bp sized

DNA. The soluble fraction containing chromatin digested to free mononucleosomes was applied to anti-HA magnetic resin pre-equilibrated in Buffer 500 and nutated overnight at 4°C. The unbound fraction was removed, and the resin washed five times with Buffer 500 to remove non-specifically bound proteins. Finally, HA-tagged proteins were eluted by boiling in 3× Laemmli buffer for 3 min. The eluate was supplemented with 50 mM DTT after aspiration from the resin. Input and elution samples were analyzed by 15% SDS-PAGE and transferred to PVDF membranes for western blotting.

## Histone methyltransferase assays

Nucleosome (0.5 µM) and COMPASS CM or COMPASS eCM (1 µM) were incubated together in 20 mM HEPES, pH 7.5, 100 mM NaCl, 1 mM DTT, 0.2 mM SAM for 30 min at 30°C. The reaction was quenched in SDS-PAGE loading buffer and analyzed by western blot (*Hsu et al., 2019*).

## Western blots

Western blots were performed in modified Towbin buffer consisting of 25 mM Tris, 192 mM glycine, 4 mg/L SDS, 10% (v/v) methanol, and proteins were transferred onto PVDF membrane at 35 V for 2 hr, on ice. Membranes were blocked in 5% (w/v) non-fat milk powder in phosphate-buffered saline (PBS) for 1 hr at 25°C before incubating overnight in diluted primary antibody in 5% (w/v) non-fat milk powder in PBST (PBS containing 0.05% (v/v) Tween-20) at 4°C. Overnight incubated membranes were washed in PBST before incubating with IR-dye conjugated secondary antibody or HRP-conjugated secondary antibody in 5% (w/v) non-fat milk powder containing PBST for 1 hr at 25°C. After incubation with secondary antibodies, the membranes were washed first in PBST and then PBS before scanning on a Li-COR Biosciences (Lincoln, NE) Odyssey IR scanner or developing with ECL reagents.

## Native chromatin immunoprecipitation for high-throughput sequencing

ChIP-seq experiments were performed using minor modifications of previously described methods (*Hawkins et al., 2010*). Briefly, doxycycline-induced, or uninduced, Flp-In T-REx 293 cells were resuspended in 10 mM HEPES, pH 7.5, 100 mM NaCl, 300 mM sucrose, 3 mM MgCl$_2$, 10 mM NEM, 0.5% Triton X-100, 1× cOmplete protease inhibitor cocktail. The suspension was incubated on ice for 2 min then centrifuged at 2500× *g* for 5 min at 4°C to pellet nuclei. Nuclei were taken up in a buffer consisting of 50 mM Tris, pH 8.0, 150 mM NaCl, 5 mM CaCl$_2$, 10 mM NEM, 1% Triton X-100 and 1× cOmplete protease inhibitor cocktail. Chromatin was digested to mononucleosomes with 10 units of MNase (Thermo Fisher, Catalog number 88216) for 15 min at 37°C. Digestion was stopped by the addition of 1 mM EDTA and the reaction mixture clarified by passing through a 25 G needle 10 times then centrifugation at 17,200× *g* for 10 min at 4°C in a microcentrifuge; 15 µg of chromatin was incubated overnight at 4°C with α-HA antibody (Sigma, Catalog number H9658) or α-H3K4me3 antibody (Active Motif, Catalog number 39159) coated Dynabeads (Thermo Fisher). Bound chromatin was washed using RIPA buffer with 500 mM LiCl for five times. Bound chromatin was eluted in TE buffer and RNA was removed by incubation with RNAseA for 30 min at 37°C followed by Proteinase K treatment at 65°C for 30 min. DNA was purified using phenol:chloroform:isoamyl alcohol. Finally, precipitated DNA was further washed three times with 80% aqueous ethanol and eluted in nuclease-free water for use in library preparation.

## Library preparation

Library preparation was performed using NEBNext Ultra II DNA Library Prep Kit from Illumina according to the manufacturer's protocol. Final libraries were amplified using KAPA HiFi Hotstart ready mix then purified and size selected for 300–600 bp final library product using AMPURE beads (Beckman Coulter). Final library quantification was performed using Qubit dsDNA HS kit and average size was determined using high sensitivity BioAnalyzer DNA kit (Agilent). Libraries were sequenced as single-end 75 cycles on the Illumina Nextseq 550 platform. Experiments were performed in duplicate.

## Sequencing data analysis

Reads were demultiplexed using the bcl2fastq tool. Raw reads were trimmed for low quality (qscore <30) and adapters using trim-galore (*Krueger et al., 2021*). Trimmed reads were mapped to hg19 using Bowtie2 (*Langmead and Salzberg, 2012*). RPKM normalization or input normalization

using BamCompare in deepTools was used to normalize signal intensity (*Ramírez et al., 2016*). Signal enrichment visualization was performed using the plotProfile function in deepTools.

## Fluorography

After SDS-PAGE separation of methylation assay components, gels were soaked in Amplify solution (GE Healthcare, Chicago, IL) for 30 min before drying on a vacuum air dryer. Dried gels were exposed to X-ray film for 1 week at –80°C. Images were developed and fixed using Kodak (Rochester, NY) GBX solutions.

## Chromatin assembly and MNase digestion

Chromatin assembly and MNase analysis proceeded essentially as described previously (*Shimada et al., 2019*). Briefly, octamer and hNAP-1 were incubated together on ice for 15 min before the addition of hTopo1, dACF1/ISWI, relaxed circular plasmid DNA, and ATP-Mg mix (0.5 M creatine phosphate, 0.5 M ATP, 1 M MgCl$_2$, 5 mg/mL creatine kinase) and incubated at 30°C for 2 hr. MNase (TaKaRa Bio Inc, Shiga, JP) was added to chromatin and digested at 25°C for 10 min. DNA was purified from the digested chromatin by miniprep kit and analyzed by 1.25% agarose gel stained with ethidium bromide.

## In vitro transcription assay

Assays were conducted as outlined in *Figure 2A* following previously described protocols (*Shimada et al., 2019*; *An and Roeder, 2004*). Gal4-VP16 (30 ng), chromatin template (25 ng), p300 (10 ng), and acetyl-CoA (5 µM) were incubated in 20 mM HEPES-KOH, pH 7.9, 1 mM EDTA, 100 mM KCl, 10% (v/v) glycerol, 0.2 mg/mL BSA at 30°C for 10 min. Preinitiation complex formation was initiated by the addition of HeLa extract (30 µg total protein) with continued incubation at 30°C for 10 min. Transcription was initiated by the addition of ATP, UTP, CTP, 3'-*O*-methyl-GTP, and [α-$^{32}$P]-CTP followed by incubation at 30°C for 1 hr. Assay products were resolved by agarose gel and analyzed by autoradiography.

## In-gel desumoylation and propionylation of lysine residues

Histone modification analysis by mass spectrometry was conducted as previously described with one key modification (*Sidoli and Garcia, 2017*). Histones were separated by SDS-PAGE and stained in Coomassie stain consisting of 0.05% (w/v) Coomassie brilliant blue dissolved in 45:45:10 (v/v) methanol/H$_2$O/acetic acid. Bands were mostly destained in 45:45:10 (v/v) methanol/H$_2$O/acetic acid and specific histone bands were excised from the gel and diced into 1 mm cubes. Gel pieces were sequentially washed in 100 mM NH$_4$HCO$_3$, pH 8.0, and then dehydrated in sufficient acetonitrile to cover the gel pieces with gentle agitation for 15 min at 25°C. Gel pieces were dried by lyophilization before rehydrating in the presence of cat.SENP2 (1 µg/µL) in HAT assay buffer on ice for 30 min, and further incubated overnight at 30°C. After desumoylation, the gel pieces were sequentially washed in 100 mM NH$_4$HCO$_3$, pH 8.0, and then dehydrated in sufficient acetonitrile to cover the gel pieces with gentle agitation for 15 min at 25°C. Histones were propionylated in-gel with 1:2 (v/v) 100 mM NH$_4$HCO$_3$/propionic anhydride for 20 min with shaking at 25°C. The propionylation reagent was aspirated away from gel pieces and they were sequentially washed in 100 mM NH$_4$HCO$_3$, pH 8.0, and then dehydrated in sufficient acetonitrile to cover the gel pieces with gentle agitation for 15 min at 25°C. In-gel propionylation was undertaken thrice to ensure complete reaction.

## In-gel tryptic digestion and peptide extraction

Histones were digested by rehydrating gel pieces with 12.5 ng/µL trypsin (Pierce) in 50 mM NH$_4$HCO$_3$ on ice for 30 min, and then incubation at 30°C overnight. The bicarbonate solution after overnight digestion contained histone peptides that were transferred into a new empty tube. Peptides were further extracted from the trypsinized gel pieces by the sequential addition of 20 µL milliQ pure H$_2$O and shaking for 15 min, followed by incubation with 20 µL acetonitrile and shaking for 15 min. All three solutions – bicarbonate, water, and acetonitrile – were pooled together. The peptide extraction process was repeated once and the combined peptide solutions were lyophilized to dryness. The dry tryptic peptides were propionylated to cap newly formed N-termini by resuspending in 30 µL 100 mM NH$_4$HCO$_3$ and adjusting to pH 8.0 before the addition of 15 µL 1:3 (v/v) propionic anhydride:ACN. The pH was quickly readjusted to pH 8.0 with NH$_4$OH before shaking for 15 min at 25°C. The reaction

mixture was dried on the lyophilizer and the propionylation of N-termini was repeated once to ensure complete reaction.

## C18 Zip-tip desalting of histone peptides

The dried histone peptides were dissolved in 50 µL 0.1% (v/v) TFA in milliQ pure $H_2O$ by vortexing and sonication in a water-bath, and the pH was adjusted to ~4.0 with TFA prior to desalting with C18 Zip-tips (EMD Millipore). The Zip-tip was first washed sequentially with 3 × 10 µL acetonitrile and then washed with 3 × 10 µL 0.1% (v/v) aqueous TFA. Peptides were bound to the resin by pipetting the peptide solution through the Zip-tip resin in 10 µL aliquots until all the solution had passed through the resin at least once. The Zip-tip resin was then washed by stepwise transfer of 5 × 10 µL of 0.1% (v/v) aqueous TFA into waste. The solid-phase bound peptides were then eluted into a clean empty tube by the stepwise transfer of 5 × 10 µL of 75% (v/v) acetonitrile in water containing 0.5 % (v/v) acetic acid. The combined eluted peptide solutions were lyophilized to dryness.

## Nano-LC and MS-MS analysis of histone peptides

The lyophilized histone tryptic peptides were dissolved in 15 µL of 10% (v/v) acetonitrile and 1% (v/v) formic acid in milliQ pure water by vortexing and sonication in a water-bath. Samples were loaded onto a C18 trap column (LC Packing) and then into a capillary column using an Agilent 1100 Series capillary LC system (Santa Clara, CA). The capillary column was packed in-house with Phenomenex C12 Jupiter resin. The mobile phase used consisted of A (0.6% (v/v) acetic acid in milliQ pure water) and B (0.6% (v/v) acetic acid in acetonitrile). Peptides were separated over a 70 min gradient from 2% to 98% Buffer B, and directly eluted through the nano electrospray ionization source into a Thermo Finnigan LTQ XL mass spectrometer (San Jose, CA) for MS and MS-MS analysis. Peptide ions were identified and analyzed manually using the Thermo Xcalibur Qual Browser. Precursor fragments were identified over at least two peptide spectra, when possible, with an error of ±0.8 Da. Mass spectrometer parameters were: Isolation width: 1 m/z; normalized collision energy: 35.0; scan range m/z: 300.00–2000.00. Dynamic exclusion was enabled.

## Acknowledgements

The authors thank the Departments of Chemistry, Genome Sciences and Pharmacology at the University of Washington and The Rockefeller University for generous support. Ning Zheng is an Investigator of the Howard Hughes Medical Institute. This research was supported by funding from NIH grants to CC (R01GM110430), RDH (R01AR065952 and R01DK103667), NZ (R01HD097408) and RGR (R01CA234561 and R01DK071900). CAL was supported by a Molecular Biophysics Training Grant T32GM008268. CEW was supported by an ARCS Foundation Fellowship and an NSF Graduate Research Fellowship (grant number DGH-1256082). CC thanks Dr Dustin Maly for assistance with stable cell-line generation.

## Additional information

### Funding

| Funder | Grant reference number | Author |
| --- | --- | --- |
| National Institutes of Health | R01CA234561 | Robert G Roeder |
| National Institutes of Health | R01DK071900 | Robert G Roeder |
| National Institutes of Health | R01GM110430 | Champak Chatterjee |
| National Institutes of Health | T32GM008268 | Calvin Jon A Leonen |
| National Science Foundation | DGH-1256082 | Caroline E Weller |

| Funder | Grant reference number | Author |
|---|---|---|
| National Institutes of Health | R01HD097408 | Ning Zheng |
| National Institutes of Health | R01DK103667 | R David Hawkins |
| National Institutes of Health | R01AR065952 | R David Hawkins |

The funders had no role in study design, data collection and interpretation, or the decision to submit the work for publication.

## Author contributions

Calvin Jon A Leonen, Conceptualization, Formal analysis, Investigation, Methodology, Resources, Validation, Writing - original draft, Writing – review and editing; Miho Shimada, Tomoyoshi Nakadai, Peter L Hsu, Patrick MM Shelton, Formal analysis, Investigation, Methodology, Resources, Validation, Writing – review and editing; Caroline E Weller, Conceptualization, Formal analysis, Investigation, Methodology, Validation, Writing – review and editing; Elizabeth L Tyson, Conceptualization, Data curation, Formal analysis, Investigation, Methodology, Validation, Visualization, Writing – review and editing; Arpit Mishra, Conceptualization, Data curation, Formal analysis, Investigation, Methodology; Martin Sadilek, Formal analysis, Methodology, Resources, Supervision, Visualization, Writing – review and editing; R David Hawkins, Conceptualization, Data curation, Formal analysis, Investigation, Methodology, Resources, Software, Supervision, Visualization, Writing – review and editing; Ning Zheng, Funding acquisition, Investigation, Methodology, Resources, Supervision, Validation, Writing – review and editing; Robert G Roeder, Conceptualization, Funding acquisition, Investigation, Methodology, Resources, Supervision, Validation, Writing – review and editing; Champak Chatterjee, Conceptualization, Formal analysis, Funding acquisition, Methodology, Project administration, Resources, Supervision, Validation, Visualization, Writing - original draft, Writing – review and editing

## Author ORCIDs

Calvin Jon A Leonen  http://orcid.org/0000-0003-3003-9021
R David Hawkins  http://orcid.org/0000-0002-2997-9457
Robert G Roeder  http://orcid.org/0000-0003-3865-8572
Champak Chatterjee  http://orcid.org/0000-0001-5772-4438

## Decision letter and Author response

Decision letter https://doi.org/10.7554/eLife.67952.sa1
Author response https://doi.org/10.7554/eLife.67952.sa2

# Additional files

## Supplementary files

• Transparent reporting form

## Data availability

All data generated or analyzed during this study are included in the manuscript and supporting files.

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
