## [Editor Report]

This manuscript uses chemically synthesized histone H4 with K12 sumoylation coupled with in vitro transcription assay and other in vitro as well as cellular biochemical assays to provide insights about the function of histone H4 K12 sumoylation. The results suggest that H4 K12 sumoylation suppresses H4 acetylation and H3 K4 methylation, two modifications that promote transcription. The biochemistry is very nicely done and the finding may be of interest to the people interested in histone modifications.

---

## [Decision Letter]

**Decision letter after peer review:**

Thank you for submitting your article "Sumoylation of the human histone H4 tail inhibits p300-mediated transcription by RNA polymerase II in cellular extracts" for consideration by *eLife*. Your article has been reviewed by 2 peer reviewers, including Hening Lin as Reviewing Editor and Reviewer #1, and the evaluation has been overseen by Cynthia Wolberger as the Senior Editor.

Essential revisions:

1) Carry out ChIP assay to obtain evidence that the effect on acetylation and methylation is also true in cells as suggested by Reviewer 2.

2) If further experiments to support the effects on H4 acetylation (comments from reviewer 1) are difficult, please revise the text and tune down the claim on acetylation.

*Reviewer #1:*

Histone H4 sumoylation is known to repress transcription but the mechanistic understanding of how sumoylation achieve this function is not well understood. The authors of this manuscript aims to use in vitro biochemical reconstitution to provide insights into this. They chemically synthesized histone H4 with K12 sumoylation and then reconsitituted nucleosomes or chromatinized plasmid with either unmodified H4 or sumoylated H4 and use that to examine the effect on H4 K12 sumoylation on p300 catalyzed acetylation, Set1-containing complex (eCM) -catalyzed H3 K4 methylation, as well as in vitro transcription. The author's data suggest that H4 K12 sumoylation suppresses H4 acetylation (but not other histone proteins' acetylation), H3 K4 methylation, and the in vitro transcription of chromatinized plasmid. Through these results, the authors have largely achieved their goal of providing mechanistic understandings for the transcription repression role of histone sumoylation.

The work has a number of strengths. The use of H4 with precisely controlled modification coupled with in vitro transcription assay is a very powerful approach. This work is technically challenging and the collaborative efforts of the two labs (Drs. Chatterjee and Roeder) are critical. The biochemical assays are nicely designed and carried out. For example, the in vitro transcription assay combined with the detection of histone acetylation in the reaction system is a really nice design. The results clearly showed that H4 K12 sumoylation suppresses transcription in the in vitro transcription assay and also inhibits H3 K4 methylation, which is consistent with the known effects of histone sumoylation.

The limitation of the work mainly lies in the significance of the acetylation result. The H4 K12 sumoylation's effect an acetylation (Figure 3) initially looked very impressive as it showed that K12 sumoylation dramatically suppressed H4 acetylation on K4, K8, K12, and K16. However, realizing that K12 sumoylation naturally blocks one of the acetylated K residue, this result becomes less impressive. In fact, I worry that if we remove K12 acetylation from the equation, the effect of sumoylation on H4 acetylation may be minor. Thus, whether the role of H4 sumoylation in transcription suppression is coming from suppressing acetylation is not very clear now.

As I mentioned above, my major concern is on the role of H4 K12 sumoylation on p300-catalyzed acetylation. The first time I read the manuscript, I was very excited about the result shown in Figure 2 and 3, but when I read it the second time, I realized that K12 is blocked by sumoylation and thus the maximal number of acetylation sites available at the N-terminal is three instead of four. Considering this, the acetylation result in Figure 3 becomes less exciting. This also raised the question whether the obvious transcription suppression effect shown in Figure 2 is really due to lack of acetylation on H4 K12 and K16. Unfortunately, to answer this question, more experiments need to be done. For example, the authors may need to prepare H4 with K12 and K14 to R mutation to block acetylation and see whether that's enough to suppress transcription in the in vitro transcription system.

*Reviewer #2:*

In this work, this manuscript describes the effects of histone H4 sumoylation on the other histone modifications, H3K4 methylation and H4 acetylation. Using in vitro synthesized H4K12su, they revealed that this modification is able to repress histone acetyltransferase p300-mediated H4 tail acetylation and Set1/COMPASS histone methyltransferase complex-mediated H3K4 methylation in vitro. Thus, their results suggest a new model for histone crosstalk of H4 sumoylation with other histone modifications. Although the analysis is solid, their research is almost only focused on in vitro study and doesn't offer any mechanism study for how histone sumoylation affects other histone modifications in vivo. Therefore, I do have a few concerns that need to be addressed, as well as several other issues to help the authors improve their manuscript.

I recommend authors perform ChIP or immunoblotting in H4K12 mutant in vivo.

Specific points of concern or questions:

1. What it the effect of histone sumoylation on H3K4 methylation and H4 acetylation in vivo? To authors need to do in vivo study to determine the correlation between these histone modifications.

2. What is the function of this crosstalk including H4K12 sumoylation in the cell? Trnascriptoin repression? They have not provided any experimental results for this question.

3. Chatterjee's group previously reported H4K12su stimulates LSD1-mediated H3K4 demethylation (Dhall, et al., 2017. ACS Chem Biol), while H4K12su inhibits the installation of H3K4 methylation by COMPASS in this manuscript. Author should reveal what is true in vivo.

4. Figure1A-D explaining how they synthesize H4K12su in octamer and mononucleosome is not novel, because Chatterjee's group already reported and showed clearly these steps in the paper (Dhall, et al., 2017. ACS Chem Biol 12, 2275-2280.). Therefore, Figure1A-D is not required for this paper.

5. Although the most well-known function of histone sumoylation is repression on the transcription, there are lots of roles of histone sumoylation in the cell. Authors should be explicit about more diverse roles including transcription activation, centromere control or DSB repair in introduction and abstract parts.

6. "in vitro" and "in vivo" needs to be represented as italic letters in the manuscript. This is error in multiple places.

---

## [Author Response]

Essential revisions:1) Carry out ChIP assay to obtain evidence that the effect on acetylation and methylation is also true in cells as suggested by Reviewer 2.2) If further experiments to support the effects on H4 acetylation (comments from reviewer 1) are difficult, please revise the text and tune down the claim on acetylation.

We did ChIP and ChIP-seq to show that the relationship between acetylation, methylation and sumoylation is observable in cells. We also revised the text to tone down the claim on acetylation in the Discussion section.

Reviewer #1:[…] As I mentioned above, my major concern is on the role of H4 K12 sumoylation on p300-catalyzed acetylation. The first time I read the manuscript, I was very excited about the result shown in Figure 2 and 3, but when I read it the second time, I realized that K12 is blocked by sumoylation and thus the maximal number of acetylation sites available at the N-terminal is three instead of four. Considering this, the acetylation result in Figure 3 becomes less exciting. This also raised the question whether the obvious transcription suppression effect shown in Figure 2 is really due to lack of acetylation on H4 K12 and K16. Unfortunately, to answer this question, more experiments need to be done. For example, the authors may need to prepare H4 with K12 and K14 to R mutation to block acetylation and see whether that's enough to suppress transcription in the in vitro transcription system.

We thank Reviewer 1 for their enthusiasm of our mechanistic work and agree with the comments. Indeed, we had written that sumoylation may repress transcription, in-part, by negative crosstalk with histone acetylation. Other mechanisms are indeed possible, such as the inhibition of histone methylation that we also showed, and cannot be excluded as contributing to transcriptional repression by SUMO without extensive investigation. Ultimately, we do not believe that these mechanisms need be mutually exclusive and they may serve to reinforce each other.

While the negative crosstalk with acetylation at 2 sites, K12 and K16, may appear somewhat unimpressive in the context of 4 total sites in the H4 tail, it is worth noting that H4K16 acetylation has the most dramatic effect on chromatin structural transitions (Shogren-Knaak, M.A. et al. *Science* 2006, *311,* 844) and is a privileged site in transcription (Dion, M.F. et al. PNAS 2005, *102,* 5501). In *Drosophila*, where males exhibit a two-fold increased transcription of their single X-chromosome (dosage compensation), a single H4K16R mutation is absolutely lethal (Copur, O. et al. PNAS 2018, *115,* 13336). This is due to the key role of H4K16ac in the activation of X-linked genes by the H4K16 acetyltransferase MOF (Akhtar, A. et al. Mol. Cell 2000, *5,* 367).

Because testing the mutants suggested by Reviewer 1 requires challenging radioactivity-based assays, we have further toned down the emphasis on acetylation in the Discussion section and made it clear that additional mechanisms of repression cannot be ruled out at this stage.

Reviewer #2:In this work, this manuscript describes the effects of histone H4 sumoylation on the other histone modifications, H3K4 methylation and H4 acetylation. Using in vitro synthesized H4K12su, they revealed that this modification is able to repress histone acetyltransferase p300-mediated H4 tail acetylation and Set1/COMPASS histone methyltransferase complex-mediated H3K4 methylation in vitro. Thus, their results suggest a new model for histone crosstalk of H4 sumoylation with other histone modifications. Although the analysis is solid, their research is almost only focused on in vitro study and doesn't offer any mechanism study for how histone sumoylation affects other histone modifications in vivo. Therefore, I do have a few concerns that need to be addressed, as well as several other issues to help the authors improve their manuscript.I recommend authors perform ChIP or immunoblotting in H4K12 mutant in vivo.

In response to the reviewer’s request we did both ChIP followed by immunoblotting from transiently transfected HEK293T cells, and also ChIP-seq experiments after generating stable HEK293T lines expressing a single copy of a SUMO-H4 fusion protein. We note that H4K12 is also a site for acetylation and methylation, hence mutation of this site would not reveal the role of sumoylation by itself, particularly when the specific SUMO ligase is unknown and cannot be knocked down/overexpressed. In fact, the steep challenge of drawing specific conclusions regarding the roles for H4K12 sumoylation in cells from any site-directed mutants is what compelled us to test the direct effect of H4K12 sumoylation in a well-defined in vitro system with nuclear extracts and semisynthetic histones.

Specific points of concern or questions:1. What it the effect of histone sumoylation on H3K4 methylation and H4 acetylation in vivo? To authors need to do in vivo study to determine the correlation between these histone modifications.

We now show by ChIP immunoblot that the negative biochemical crosstalk between H4 sumoylation and H4K16 acetylation is retained when averaged across the entire Micrococcal nuclease-accessible chromatin from HEK293T cells. This is not surprising as the negative crosstalk occurs *in cis* within the same H4 tail. However, H3K4me3 immunoblots with the same ChIP samples did not reveal significant negative biochemical crosstalk *in trans* between H4 sumoylation and H3K4me3. This is likely due to the fact that MLL-family methyltransferases do not contain analogs of the Spp1 subunit of COMPASS that enforces negative crosstalk with H4 sumoylation as we showed in the manuscript. Moreove, H3K4me3 not only marks actively transcribed genes but also occurs in broad domains on either side of the transcription start sites (Benayoun, B.A. et al. Cell 2014, *158,* 673 and Liu, X. et al. Nature 2016, *537,* 558) where different enzyme complexes may install it. It is noteworthy that COMPASS is the only H3K4 methyltransferase in yeast, while humans have the SET1 and MLL family of enzymes. The human SET1 complex does have an analog of the Spp1 subunit (CFP1). Therefore, based on the localization of CFP1, and thereby SET1, to actively transcribed gene promoters (Brown, D.A. et al. Cell Reports 2017, *20,* 2313) we used ChIP-seq to interrogate the co-occupancy of H3K4me3 and H4 sumoylation between transcription start and end-site in HEK293T cells in which SUMO-H4 was stochastically incorporated at detectable levels. This analysis revealed the unambiguous negative correlation between H3K4me3 and SUMO-H4 occupancy between the transcription start and end sites of genes. We have added these new data in the revised manuscript.

2. What is the function of this crosstalk including H4K12 sumoylation in the cell? Trnascriptoin repression? They have not provided any experimental results for this question.

Correlative evidence from the expression of SUMO-H4 fusions, which mimic H4 tail sumoylation, leading to gene repression in cells was shown by Shelley Berger’s labs in Genes Dev. 2006, *20,* 966. Histone H4 sumoylation was generally associated with gene repression by localizing the SUMO ligase Ubc9 near gene promoters (Shiio and Eisenman, PNAS 2003, *100,* 13225), but there was absolutely nothing known about the specific and direct effect of H4K12 sumoylation on transcription until our study. This is because H4K12 sumoylation is not easily accessible in cells due to the lack of knowledge regarding (1) its specific ligase, (2) the timing of this modification, and (3) the lack of antibodies that specifically recognize H4K12su.

3. Chatterjee's group previously reported H4K12su stimulates LSD1-mediated H3K4 demethylation (Dhall, et al., 2017. ACS Chem Biol), while H4K12su inhibits the installation of H3K4 methylation by COMPASS in this manuscript. Author should reveal what is true in vivo.

Both effects, the stimulation of LSD1 activity at methylated H3K4 (our previous work) and the inhibition of COMPASS-mediated methylation at H3K4 (this work) may be expected to lead to the same outcome – repression of gene transcription. However, we note that only a subset of neuronal genes in non-neuronal cells are targeted for repression by the REST1 co-repressor CoREST1, and that a further subset of these neuronal genes are actually regulated by SUMO (Ouyang, J. et al. Mol Cell 2009**,**
*34,* 145). Therefore, LSD1 stimulation by SUMO would be observed at very specific genes as shown by Ouyang et al. in 2009. Both the timing and genomic location of histone sumoylation, and the co-occurrence of specific methyltransferases/demethylases would dictate whether SUMO acts by preventing H3K4 methylation, or acts to recruit CoREST1 and therefore LSD1 to methylated regions of chromatin.

4. Figure1A-D explaining how they synthesize H4K12su in octamer and mononucleosome is not novel, because Chatterjee's group already reported and showed clearly these steps in the paper (Dhall, et al., 2017. ACS Chem Biol 12, 2275-2280.). Therefore, Figure1A-D is not required for this paper.

We prefer to retain Figure 1A as it introduces a complex multi-step semisynthesis to a broad range of readers in a highly abbreviated single-line form while showing the key ligation auxiliary. The semisynthesis of H4K12su is a linchpin of the entire manuscript and it is critical to show Figure 1B, which indicates excellent purity of the H4K12su species that enabled tandem-MS analysis of acetylation sites. These are entirely unpublished spectra specific to reagents used in this manuscript. Figures 1C and 1D relate to octamers that were not used in the previous publication the reviewer is referring to. Those octamers were methylated, whereas these are not, and hence the nucleosomes too are different.

5. Although the most well-known function of histone sumoylation is repression on the transcription, there are lots of roles of histone sumoylation in the cell. Authors should be explicit about more diverse roles including transcription activation, centromere control or DSB repair in introduction and abstract parts.

We have indicated potential roles of histone sumoylation in other processes to both Abstract and Introduction sections of the manuscript and cited a review from Ryu and Hochstrasser.

6. "in vitro" and "in vivo" needs to be represented as italic letters in the manuscript. This is error in multiple places.

We have made this change based on the reviewer’s request.

Finally, we thank Reviewer 2 for the questions/suggestions.